# Polyvinyl chloride-based dielectric elastomer with high permittivity and low viscoelasticity for actuation and sensing

Jianjian Huang[1], Xiaodie Zhang[1], Ruixue Liu[1], Yonghui Ding [2]✉ & Dongjie Guo [1]✉

Dielectric elastomers (DEs) are widely used in soft actuation and sensing. Current DE actuators require high driving electrical fields because of their low permittivity. Most of DE actuators and sensors suffer from high viscoelastic effects, leading to high mechanical loss and large shifts of signals. This study demonstrates a valuable strategy to produce polyvinyl chloride (PVC)-based elastomers with high permittivity and low viscoelasticity. The introduction of cyanoethyl cellulose (CEC) into plasticized PVC gel (PVCg) not only confers a high dielectric permittivity (18.9@1 kHz) but also significantly mitigates their viscoelastic effects with a low mechanical loss (0.04@1 Hz). The CEC/PVCg actuators demonstrate higher actuation performances over the existing DE actuators under low electrical fields and show marginal displacement shifts (7.78%) compared to VHB 4910 (136.09%). The CEC/PVCg sensors display high sensitivity, fast response, and limited signal drifts, enabling their faithful monitoring of multiple human motions.

Dielectric elastomers (DEs) are emerging as promising materials for actuation and sensing. DE device, usually composed of a thin elastomeric film sandwiched between two compliant electrodes, is able to transduce electrical to mechanical energy as actuators[1–3], and vice versa as sensors[4–6]. DE actuators (DEAs) are attractive artificial muscles due to their high energy density[7] and conversion efficiency[8], and fast response[9]. DE sensors (DESs), such as electronic skin, exhibit outstanding advantages, including high flexibility, low weight, broad linear response range as compared to the stiff inorganic sensors, e.g., piezoelectric and magnetostrictive sensors[10]. However, currently existing DEs suffer from an intrinsically low dielectric permittivity ($\varepsilon = 2-10$). For example, the dielectric permittivity of widely used DEs are 2.2–3.0 for polydimethylsiloxane (PDMS)[11], 4.4–4.7 for VHB acrylic elastomer (3 M)[12,13], and 4.0 for pure polyvinyl chloride (PVC)[14]. DE actuators with low dielectric permittivity, such as PDMS and VHB materials, often require high driving electric fields (>20 V/μm) to achieve large actuations, which would lead to the high risks of current leakage[15] and electrical breakdown[16] when such high driving electric fields are close to their breakdown strength. In addition, a high value of several kilovolts arises safety issues and brings about the problem of using a bulky high-voltage power supply system[17].

Numerous efforts have been devoted to increase the dielectric permittivity and mechanical flexibility to generate a large actuation under relatively low driving voltages[11–13,16–19]. For instance, the seminal work from Kofod's group enhanced the relative permittivity of the PDMS elastomer from 3.0 to 5.9 and decreased the elastic modulus from 1900 to 550 kPa by grafting small molecules with high dipole moment to the elastomer matrix, leading to significant improvement of their electromechanical performances[19]. In addition, the reduction of the film thickness is an alternative method to improve the actuation performance[20–23]. For example, Shea and his co-workers demonstrated that the actuation strain of 7.5% could be generated with a 3 μm thick film under a driving voltage of 245 V[20]. By contrast, it required much higher driving voltage of 3.3 kV to generate the same actuation strain with the 30 μm thick film. Despite these positive outcomes, thin film actuators often require complicated fabrication processes and are associated with high prevalence of an electromechanical instability[24].

[1]State Laboratory of Surface & Interface, Zhengzhou University of Light Industry, Zhengzhou 450002, China. [2]Center for Advanced Regenerative Engineering, Department of Biomedical Engineering, Northwestern University, Evanston, IL 60208, USA. ✉e-mail: yonghui.ding@northwestern.edu; djguo@zzuli.edu.cn

Among DEs, a plasticized polyvinyl chloride gel (PVCg) has shown a great potential for both actuation and sensing applications as they are easily available, low-cost, and electrically inactive[25–28]. The plasticizers are often introduced into PVC matrices in order to produce highly flexible PVCg elastomers with high flexibility by weakening the interaction forces among PVC chains[29] (Supplementary Fig. 1). However, the plasticized PVCg suffer from low breakdown strength and inherent strong viscoelastic effects[30], which leads to time-dependent change of internal stress and strain[31], i.e., creep (Supplementary Fig. 3). Notably, such viscoelastic effects are widely presented in other elastomers such as VHB[12,32], polyurethane (PU)[33], and polyurethane acrylate (PUA)[34]. Although the creep could be utilized to trigger different mechanisms of deformation, such as bending, contracting, and crawling, it results in evident mechanical loss, stress relaxation, and viscoelastic hysteresis, leading to instability of output signals over time as well as delayed response[35]. For instance, the creep-driven PVCg actuators[25] show more frequent jump of output signals and the delayed electromechanical response as compared to actuators that are primarily driven by the Maxwell force. The viscoelasticity of PVCg sensors often leads to a large drift of bulk permittivity and output signals over time because of the random rearrangement of polar groups of PVC chain during stretching[36]. However, the viscoelastic effects of PVCg-based DEs have been largely over-looked. The mitigation of their viscoelastic effects without compromising their electromechanical functions remains warranted.

In this study, we reported a valuable strategy to produce a PVCg-based dielectric elastomer with desired properties, i.e., high permittivity, low viscoelasticity, and high flexibility, and further demonstrated its performances in both actuation and sensing applications, especially under low driving electrical fields. Specifically, an introduction of cyanoethyl cellulose (CEC) into the PVCg matrix, i.e., CEC/PVCg results in a 2.5-fold increase in dielectric permittivity (18.9) and up to 90% reduction of viscoelastic effects as compared to PVCg. The high permittivity is resulted from the enlarged interfacial capacitance by the giant orientational polarization of the C≡N dipole moments within CEC/PVCg elastomers. The low viscoelasticity is ascribed to the presence of multiple molecular interactions, including the H-bonding[37] and electrostatic interactions among PVC, CEC, and the plasticizer, which severely hinder the free motion of PVC chain (Fig. 1). The produced CEC/PVCg elastomers exhibit high electromechanical coupling properties, high accuracy, and a long-term stability without the evident creep over at least 1000 test cycles in both actuation and sensing applications.

## Results

### Physicochemical characterization of CEC/PVCg elastomers

The propionitrile was grafted onto the commercial microcrystalline cellulose (MC) via the Michael addition reaction to enhance its electric

polarizability and affinity to the PVCg. A large amount of CEC (up to 17 wt%) was successfully incorporated into PVCg matrix with homogeneous distribution (Fig. 2a, b). The prepared CEC/PVCg films were clear and exhibited a high light transmittance of >85.5% in the wavelength range of 400–1000 nm (Supplementary Fig. 6), which was similar to pristine PVCg elastomer, suggesting the good chemical compatibility between CEC and PVCg. The presence of C≡N of CEC in Fourier transform infrared (FTIR) spectra revealed the successful Michael addition between MC and acrylonitrile (Supplementary Note 2 and Supplementary Fig. 5).

The chemical analysis indicated the successful incorporation of CEC into PVCg matrix. Specifically, the X-ray photoelectron spectroscopy (XPS) C 1s (Fig. 2c) and N 1s (Fig. 2d) spectra showed the presence of C≡N from CEC in the CEC/PVCg elastomer while C≡N was absent in PVCg. The FTIR spectrum of CEC/PVCg contained characteristic peaks of C≡N from CEC and of C-O-C and C=O from PVC (Fig. 2e). In addition, X-ray diffraction (XRD) spectra showed the partially crystallized structure of CEC with a peak at 10.4°[38] (Fig. 2f), which was vanished in the XRD spectrum of CEC/PVCg. It suggested the change of crystal structure of CEC within PVCg matrix due to the interactions between CEC and PVCg.

### Electromechanical coupling property of CEC/PVCg elastomers

A large dielectric permittivity ($\varepsilon_r$) of DE is highly desirable for both actuation and sensing applications, as it determines the allowable strain for actuation and capacitance for sensing according to the following performance figures of merit, $\mathbf{S_z} = -\varepsilon_r \varepsilon_0 \mathbf{E}^2/Y$ and $C = \varepsilon_r \varepsilon_0 A/d$. It was found that the introduction of 11 wt% CEC increased dielectric permittivity of CEC/PVCg film ($\varepsilon_r = 18.9$ at 1 kHz) by 2.5 folds as compared to the pristine PVCg film ($\varepsilon_r = 7.6$ at 1 kHz) (Fig. 3b and Supplementary Note 3). Notably, the dielectric permittivity of CEC/PVCg was at least four times higher than the PDMS ($\varepsilon_r = 2.2–3.0$)[11] and commercial 3 M VHB ($\varepsilon_r = 4.4–4.7$)[12,13] materials. The dielectric permittivity was sensitive to the loading concentration of CEC. The dielectric permittivity of CEC/PVC increased with loading concentrations in the range of 0–11 wt% because of the following three reasons (Fig. 3a): (1) the formation of significant polarization capacitance by the introduction of abundant dipole groups of C≡N[19]; (2) the generation of substantial interfacial capacitance at interfaces between CEC and PVCg due to the sharp difference between their conductivity[39], i.e., $3.14 \times 10^{-9}$ S/cm for CEC (Supplementary Fig. 5g) and $3.14 \times 10^{-10}$ S/cm for PVCg (Supplementary Fig. 8c); and (3) the augmentation of interfacial capacitance by the orientational polarization of C≡N groups under the external electrical field[40]. Interestingly, the dielectric permittivity reached the highest values with the addition of 11 wt% CEC and slightly decreased with further increase of CEC concentrations, which was likely due to the introduction of more air bubbles within the film[41].

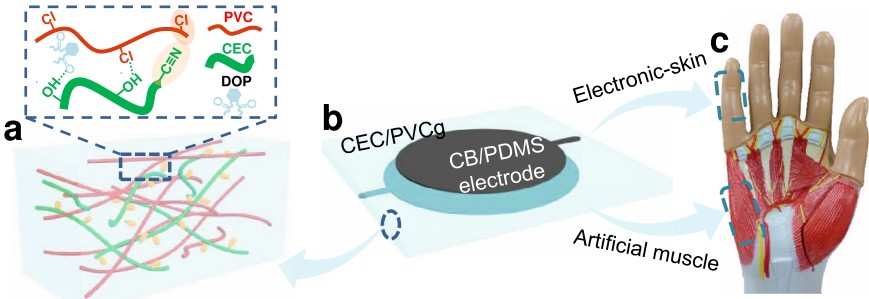

**Fig. 1 | Schematic illumination of CEC/PVCg matrix for actuation and sensing applications. a** The high permittivity of CEC/PVCg is granted by strong orientational polarization of C≡N dipole moments and giant charge accumulation at heterogeneous interfaces. The low viscoelasticity of CEC/PVCg is resulted from the multiple molecular interactions, including the H-bonding and electrostatic

interactions among PVC, CEC and the plasticizer of DOP. The CEC/PVCg elastomer is sandwiched between two layers of flexible carbon black/PDMS electrodes (**b**) to produce a parallel capacitor structure for actuation (e.g., artificial muscle) and sensing (e.g., electronic skin) applications (**c**).

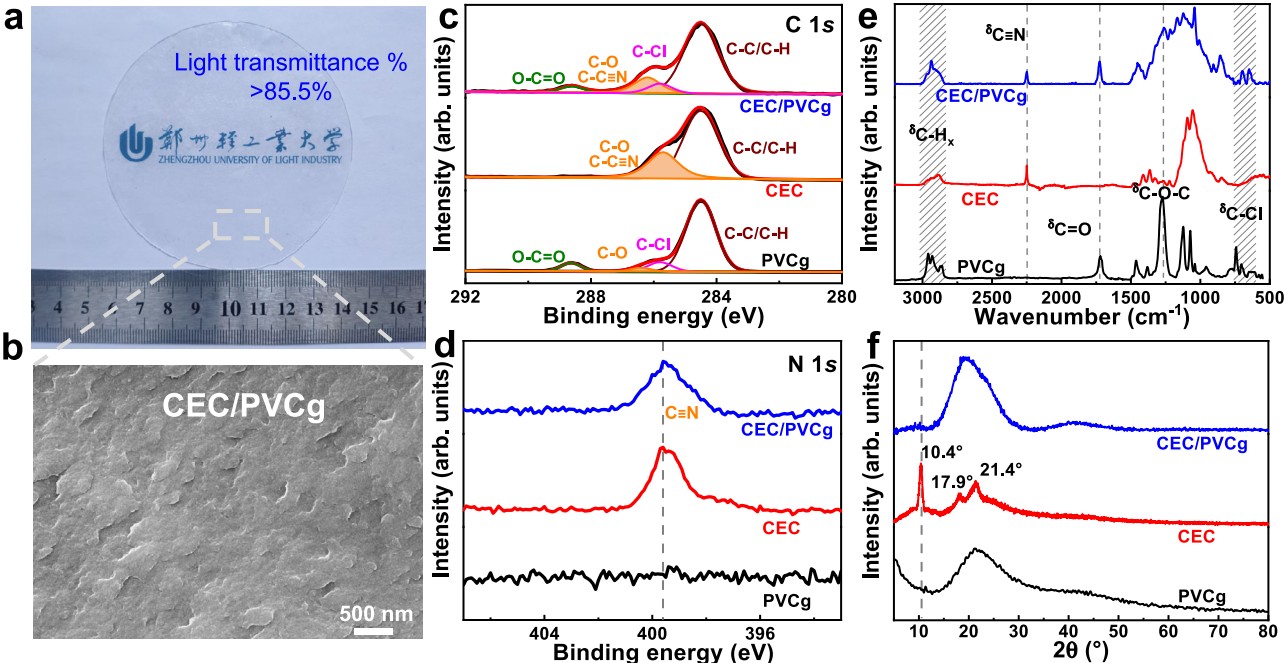

**Fig. 2 | Physicochemical characterization of CEC/PVCg.** An optical image (**a**) and cross-sectional SEM image (**b**) of the transparent CEC/PVCg elastomer. High-resolution XPS spectra of C 1*s* (**c**) and N 1*s* (**d**) regions, FTIR spectra (**e**), and XRD diffraction patterns (**f**) of PVCg, CEC, and CEC/PVCg elastomer. The dashed lines indicated the major distinction among different samples, and the shadings represent the vibration regions. The signature of C≡N was shown at 286.2 eV in XPS C 1*s* region, 399.6 eV in XPS N 1*s* region, and FTIR absorption peak at 2250 cm⁻¹.

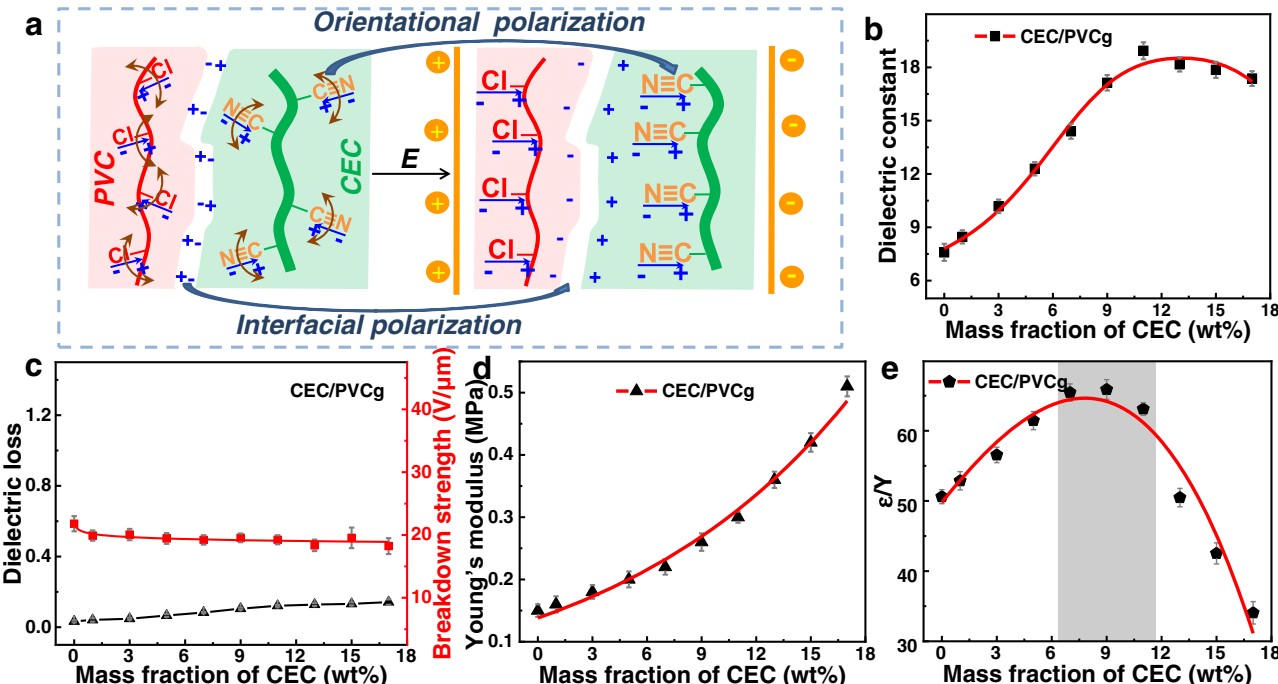

**Fig. 3 | Evaluation of CEC/PVCg elastomers' electromechanical coupling capability. a** Schematic illumination of mechanisms by which the introduction of CEC significantly increased the dielectric permittivity of CEC/PVCg. **b** Dielectric permittivity, **c** dielectric loss and breakdown strength, **d** Young's modulus, **e** electromechanical sensitivity ($k = \varepsilon/Y$) of CEC/PVCg elastomers at various loading concentrations of CEC. The red and green lines represent the PVC and CEC chains, respectively. The shading region of **e** suggests the optimal proportions for preparing elastomers. The error bars represent the standard deviations.

Dielectric loss is crucial for most DEs. Because the high levels of dielectric loss can result in substantial increases in both temperature and conductivity, which could potentially lead to thermal or electrical breakdown[42–44]. Unfortunately, the commonly used strategies of increasing dielectric permittivity, such as the addition of inorganic polar particles and conductive particles, were often associated with a substantial increase in the dielectric loss (e.g., >0.5 at 1 kHz), which would lower the breakdown strength and reduce the lifetime of

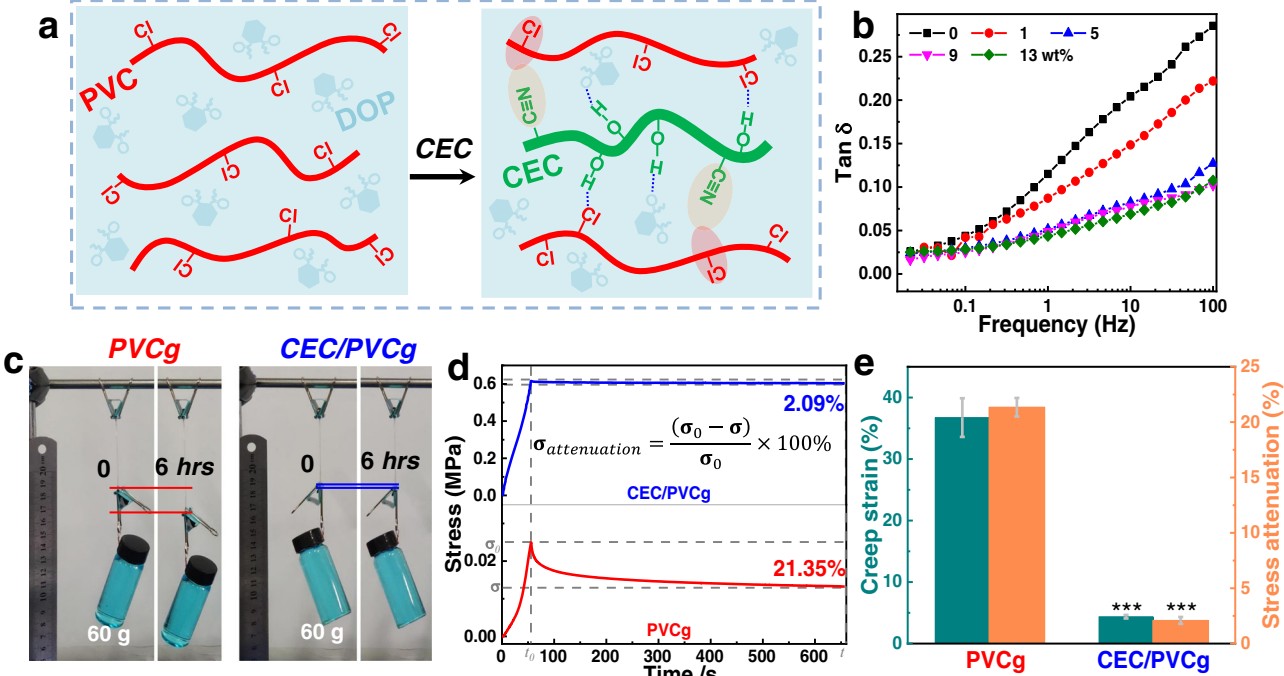

**Fig. 4 | Dynamic and static viscoelasticity of CEC/PVCg elastomers. a** Schematic illumination of the multiple molecular interactions between CEC (green) and PVC (red), which contribute to the reduced viscoelasticity of CEC/PVCg elastomers. **b** Mechanical losses (tan $\delta = G''/G'$) of CEC/PVCg elastomers with 0, 1, 5, 9, 13 wt% loading concentrations of CEC under frequencies of 0.01–100 Hz. **c** Creep-induced strains of PVCg and CEC/PVCg elastomers under a fixed load of 60 g for 6 h. **d** Stress attenuation of PVCg and CEC/PVCg elastomers where the strain was fixed at 100% for 10 min. **e** Quantifications of creep-induced strains and stress attenuation of PVCg and CEC/PVCg elastomers, where the significance is denoted by the following marks: *** for $p < 0.001$ as compared to the pristine PVCg. The dashed lines and oval shadings of **a** represent the H-bond and dipole-dipole interactions, respectively. The error bars represent the standard deviations.

actuators[45,46]. According to dielectric percolation theory, the increase of dielectric loss is mainly due to the increase of conductive filler materials and the formation of a connected conductive network. For instance, the addition of 6–9 wt% multiwalled carbon nanotube into PDMS resulted in very high dielectric losses of 3.74–4.00 @ 1.0 kHz[47]. By contrast, the introduction of CEC into PVCg matrix in this study induced a marginal increase, resulting in dielectric losses of <0.15 at 1 kHz for CEC/PVCg elastomers (Fig. 3c), which are still in a low level for DEs as suggested by previous study[48]. The limited increase of dielectric loss following the addition of CEC could be ascribed to the highly electrical insulating nature of CEC (Supplementary Note 2). The measurement of conductivity indicated that the CEC/PVCg elastomers ($4.0 \times 10^{-9}$ S/cm) were in a highly insulating state. Importantly, the breakdown strength was only slightly decreased by 10.4% following the addition of CEC and kept almost constant for CEC/PVCg elastomers with different loading concentrations of CEC (Fig. 3c and Supplementary Fig. 4). The detected breakdown strength of CEC/PVCg elastomers with 9 wt% CEC was 19.53 V/μm. Thus, a much lower driving electric field of 9.09 V/μm was used in all of experiments in this study in order to prolong the working life of device and achieve a stable performance while generating a desired large area strain.

In addition, to evaluate the energy conversion capability between electric energy and mechanical energy of the CEC/PVCg film, the electromechanical coupling sensitivity was measured and calculated according to the following performance figure of merit: $k = \varepsilon/Y$, where $k$ is the sensitivity, $\varepsilon$ is the permittivity, and $Y$ is the Young's modulus. The results from tensile test showed that the Young's modulus of PVCg was increased from 0.15 MPa up to 0.51 MPa by the addition of CEC (1–17 wt%) (Fig. 3d and Supplementary Fig. 7). The calculated electromechanical coupling sensitivity of CEC/PVCg was initially increased and then decreased as a function of CEC loading concentrations (Fig. 3e and Supplementary Table 1). The maximum sensitivity of 65.88, i.e., the highest energy

conversion efficiency was achieved at the CEC/PVCg with 9 wt% CEC, which is 3 times higher than 3 M VHB with sensitivity of 21[12].

## Viscoelasticity of CEC/PVCg elastomers

Given the percolation behavior of plasticizes and subsequent weakening of the molecular interactions (Supplementary Note 1), the existing plasticized PVCg suffers from inherent strong viscoelastic effects, resulting in a significant mechanical loss and time-dependent behaviors[29,49] (Supplementary Fig. 2 and Fig. 3). Here the introduction of multiple molecular interactions between CEC and PVCg could potentially hinder the free motion of PVC chains and thus' reduce the viscoelastic effects (Fig. 4a and Supplementary Note 3). The measurement of dynamic viscoelasticity using a rheometer showed that the mechanical loss of CEC/PVCg (i.e., tan $\delta = G''/G'$) was very limited, i.e., tan $\delta = 0.04$ at 1 Hz, which was a dramatic decrease by 67% and 96% when compared to the PVCg (tan $\delta = 0.12$) and the 3 M VHB (tan $\delta = 0.93$)[12], respectively (Fig. 4b and Supplementary Fig. 9). The static viscoelasticity was accessed by measuring the creep-induced elongation (%) under constant load and stress relaxation under constant strain. Under constant load of 60 g for 6 h, the creep-induced elongation of CEC/PVCg (4.4%) was decreased by 89% as compared to the pristine PVCg (36.7%) (Fig. 4c, e and Supplementary Fig. 10). Similarly, under constant strain of 100% for 10 min, the recorded stress attenuation of CEC/PVC was only 2.1%, which represented 90% decrease as compared to the pristine PVCg (21.4%) (Fig. 4d, e). Together, the results suggest that the introduction of CEC into PVCg has dramatically reduced their viscoelastic effects, which would benefit their actuation and sensing applications of the CEC/PVCg elastomers.

## DEA application of CEC/PVCg elastomers

DEA shows promise for applications in robotics, bionics, and other biomedical fields given their capability of producing similar stress-

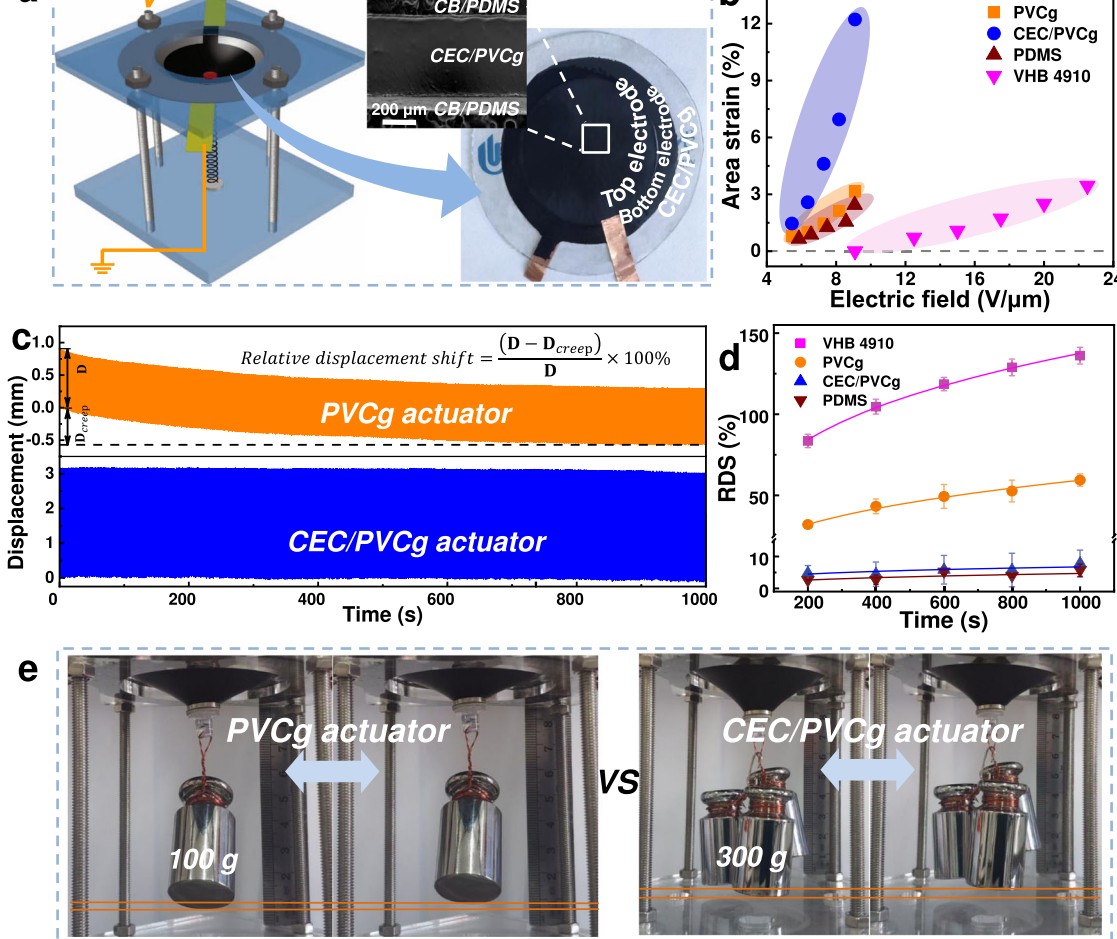

**Fig. 5 | Actuation test. a** Photograph and cross-sectional SEM image (right) of the prepared DEA device, where a CEC/PVCg film (transparent, ~470 μm thick) was sandwiched between two CB/PDMS electrodes (black, ~15 μm thick). The sandwiched structure was fixed by a circle frame (φ = 50 mm) and took out-of-plane actuation as assisted by an elastic spring (left schematic). The sheet resistance and bulk conductivity of the electrode were ~1140 Ω/sq and ~42 mS/cm, respectively. **b** The produced area strains by the PVCg, CEC/PVCg, PDMS, and VHB 4910 actuators under various driving electric fields. **c** The displacement profiles of the PVCg and CEC/PVCg actuators over 1000 cycles of actuations, i.e., 1000 s. **d** The quantified relative displacement shifts (RDS) of the PVCg, CEC/PVCg, PDMS, and VHB 4910 actuators over actuation for 200, 400, 600, 800, and 1000 s. The RDS were calculated by the following equation RDS $= \frac{|\mathbf{D} - \mathbf{D}_{creep}|}{\mathbf{D}} \times 100\%$, where **D** is the amplitude of displacement and $\mathbf{D}_{creep}$ is the shift of the displacement as shown in **c**. The error bars represent the standard deviations. **e** Weight-lifting tests to evaluate the recovery forces of PVCg and CEC/PVCg actuators.

strain behavior as natural muscle by directly converting electrical energy into mechanical work. To demonstrate the actuation performance of CEC/PVCg elastomers, a DEA was prepared by sandwiching a CEC/PVCg or PVCg film between two electrodes being made from CB/PDMS composite (Fig. 5a).

The out-of-plane displacements and strains were produced by both PVCg and CEC/PVCg actuators under an electrical field of 9.09 V/μm depending on the loading concentrations of CEC (Supplementary Note 4 and Supplementary Fig. 13). The CEC/PVCg actuators with 9 wt% CEC displayed the largest displacement, i.e., 3.23 mm (Supplementary Movie 2), which was 3.6-fold larger than the PVCg actuator, i.e., 0.90 mm (Supplementary Movie 1). According to the strain model shown in Supplementary Fig. 12, the counted area strain generated by the CEC/PVCg actuators was 12.22% (with 9 wt% CEC, under driving voltage of 9.09 V/μm, pre-strain of 25%), which represents 3.9-fold increase as compared to the PVCg actuators (Fig. 5b). In addition, we prepared the commonly used PDMS and VHB-based actuators and measured their actuation strains under the same conditions as the comparison (Fig. 5b, Supplementary Table 2, and Supplementary Fig. 15). PDMS actuators produced strains of 0.65–2.44% under electrical fields of 5.85–9.07 V/μm. VHB 4910

actuators could not be triggered, i.e., 0% area strain, under electrical fields <12.5 V/μm and produced only 0.72–3.45% strains by further increasing electrical fields to 12.5–22.5 V/μm. Therefore, our CEC/PVCg actuators generated significantly larger actuation strains, i.e., >5 times, than commonly used PDMS and VHB 4910 actuators within <22.5 V/μm electrical field, which was largely attributed to the augmentation of the electromechanical coupling sensitivity $k$ of CEC/PVCg (Fig. 3e). In addition, the amplitude of flection displacement of our CEC/PVCg actuators was decreased with the increase of the driving frequency with fast response time (0.1–0.5 s) (Supplementary Fig. 14), which represented a characteristic electromechanical behavior of Maxwell field driven actuators. By contrast, it often took 5–20 s per cycle for creep-driven actuators, such as PVCg actuators[26].

The viscoelastic drift is a major drawback of existing DEs, which is detrimental to their actuation and sensing applications. Large displacement drifts have been observed on the PVCg based actuators as well as other commonly used DEAs, including VHB[12,32], PU[33] and PUA[34] based actuators. The displacements of the PVCg, CEC/PVCg, PDMS, and VHB 4910 actuators were measured and recorded over actuation for 1000 cycles, i.e., 1000 s (Fig. 5c and

Supplementary Fig. 16). The CEC/PVCg and PDMS actuators produced remarkably stable displacement profiles. By contrast, apparent displacement shifts were observed over time for PVCg and VHB 4910 actuators. Relative displacement shifts (RDS) were calculated to quantify the viscoelastic effects (Fig. 5d). It was found that RDS values increased with time, i.e., number of actuation cycles, for VHB 4910 and PVCg actuators, while remaining almost constant for CEC/PVCg actuators. The relative shifts over 1000 cycles of CEC/PVCg actuators (7.78% of RDS) represented 87% and 94% reductions as compared to PVCg (59.40% of RDS) and VHB 4910 actuators (136.09% of RDS). PDMS actuators (5.70% of RDS) displayed similar viscoelastic drifts to CEC/PVCg actuators. The low viscoelasticity and strong electromechanical coupling capability of the CEC/PVCg elastomers could lead to highly effective energy conversion from electrical to mechanical energy with limited mechanical loss over time. In addition, the displacement curves of PVCg and CEC/PVCg actuators showed a constant strain width (i.e., wave amplitude) over at least 1000 cycles, suggesting the strong chemical affinity between the CB/PDMS electrode and the PVCg or CEC/PVCg matrix.

Moreover, the generated recovery forces, which a critical measure of actuation performance were evaluated by examining the maximum weight that the DEA can support while maintaining the periodic vibration under the pulsed electronic signal. The CEC/PVCg actuators could lift a weight of 300 g (Fig. 5e and Supplementary Movie 4), while the PVCg actuators only lifted a weight of 100 g (Supplementary Movie 3). Together, the CEC/PVCg actuators with a strong electromechanical coupling capability and low viscoelasticity produced large out-of-plane displacements, an excellent long-term stability of actuation, and high recovery forces.

## DES application of CEC/PVCg elastomers

In addition to actuation, the DE has been also used as wearable and stretchable stress/strain sensors for a variety of applications such as healthcare monitoring and human motion detection by transducing external mechanical stimuli into electrical signals. To demonstrate the sensing application of the CEC/PVCg elastomers, the periodic strain driven by a linear reciprocating actuator was applied to the prepared DES devices, including CEC/PVCg, PVCg, PDMS, and VHB 4910-based sensors (Fig. 6a and Supplementary Fig. 17). The profiles and periods of the capacitance signals (output) that were generated from both PVCg and CEC/PVCg sensors were identical to the strain signals (input) (Fig. 6b and Supplementary Movie 5), suggesting that the mechanical signal can be accurately converted into the electric signal by the prepared sensors (Supplementary Note 5). The CEC/PVCg sensors showed the fast response time, e.g., 1.0, 0.5, and 0.25 s under frequencies of 0.5-2.0 Hz (Supplementary Fig. 19). Notably, the CEC/PVCg sensors generated a significantly higher signal/noise ratio, baseline capacitance and capacitance width (i.e., $\Delta C$, difference between peak capacitance $C$ and baseline capacitance $C_O$) than the PVCg sensors because of the higher permittivity of CEC/PVCg matrix[36]. Moreover, the CEC/PVCg sensors showed the highest sensitivity ($S$) among four types of sensors that we studied here. For instance, the sensitivity of CEC/PVCg sensors was 3.1-fold, 1.5-fold, and 1.7-fold higher than PDMS, VHB 4910, and PVCg sensors, respectively, in the displacement range of 7-14 mm (Fig. 6c and Supplementary Fig. 18).

Notably, the capacitance generated by PVCg and VHB 4910 sensors showed an apparent drift over the recording time of 60 min (i.e., 1440 cycles) (Fig. 6d and Supplementary Fig. 20), which is in line with the previous report[28]. By contrast, CEC/PVCg and PDMS sensors produced remarkably stable capacitance signals without visible drift over at least 60 min, which was resulted from the low viscoelasticity and the inhibition on the rearrangement of their polar functions by the multiple molecular interactions. The relative standard deviation (RSD) of capacitances over 1440 cycles was analyzed to quantify the

stability of sensors (Fig. 6e). The results showed that our CEC/PVCg and PDMS sensors displayed much lower RSD values (5.75% and 3.67%) of relative capacitances, i.e., higher stability, than PVCg (9.84%) and VHB 4910 (8.2%) sensors. Altogether, our CEC/PVCg sensors demonstrated the superior overall sensing performances regarding of high sensitivity and stability compared to existing PVCg, PDMS, and VHB 4910 sensors.

In addition, our CEC/PVCg sensors showed a high sensitivity in a wide range of deformation, which cannot be achieved with commercially available sensors (Supplementary Table 3). The high sensitivity of existing commercial sensors has been heavily relying on the sophisticated and complex design of their structures, which requires the costly and time-consuming micro-/nano-manufacturing techniques, such as micro-electromechanical systems (MEMS). By contrast, our CEC/PVCg sensors were fabricated by a very simple and highly accessible mold-casting method with an estimated cost of ~ $ 2.00 per sensor.

The CEC/PVCg sensors were further applied to monitor the human body motions during which the capacitance signals could be collected online through a wireless transmission module and a single-chip microcomputer. The motion-induced strains were accurately reflected by the changes of recorded capacitance (Fig. 6f–h). Specifically, the changes of capacitance increased with the intensity of the motion. For instance, the elbow motion resulted in a much larger jump of capacitance due to the more intensive bending than the finger and wrist motions. Interestingly, the speed of leg motion was faithfully recorded through the changes of frequency and amplitude of the measured capacitance (Fig. 6i and Supplementary Movie 6).

## Discussion

We reported a valuable strategy to fabricate CEC/PVCg elastomers with high permittivity and low viscoelasticity. A cellulose was functionalized with cyanoethyl groups and introduced to a plasticized PVCg. The resultant CEC/PVCg elastomers presented a high permittivity (18.93 at 1 kHz), an optimal electromechanical coupling sensitivity of 65.88, and low viscoelastic effects (mechanical loss tan $\delta$ = 0.04 at 1 Hz). As a result, the CEC/PVCg actuators demonstrate higher actuation performance over the existing DE actuators, such as PDMS and VHB 4910, under low driving electrical fields. Specifically, the CEC/PVCg actuators generated a large area strain of 12.22% and high recovery force of 300 g under a low electrical field of 9.09 V/μm. The CEC/PVCg sensors exhibited a high capacitance width and sensitivity. Both CEC/PVCg actuators and sensors showed remarkable long-term stability with negligible viscoelastic drifts during at least 1000 actuation cycles and 1440 sensing cycles. In future work, we will develop a single device of CEC/PVCg with concurrent actuation and sensing functions in order to realize the so-called self-sensing mode for actuator's closed-loop control without additional stress/strain sensors. We anticipate that our strategy would benefit a variety of actuation and sensing applications such as artificial muscle, electronic sensory skin, and so on.

One limitation of current devices is the intrinsic low breakdown strength, i.e., 21.79 V/μm, of the traditional plasticized PVCg, as compared to the existing PDMS and VHB 4910. The use of higher driving electrical fields can offer high energy conversion density and energy harvesting with VHB 4910-based actuators[50]. A potential, facile solution to increase the breakdown strength and lower the required driving voltages is the reduction of film thickness of DEs[51,52]. The current film fabrication method in this study, i.e., mold casting, produced a relative thick film (~470 μm) of DEs although it is very straightforward to use. We are currently working on the fabrication of the thinner film, e.g., ~100 μm by using spray coating method, or down to ~50 μm by using spinning coating method, to increase

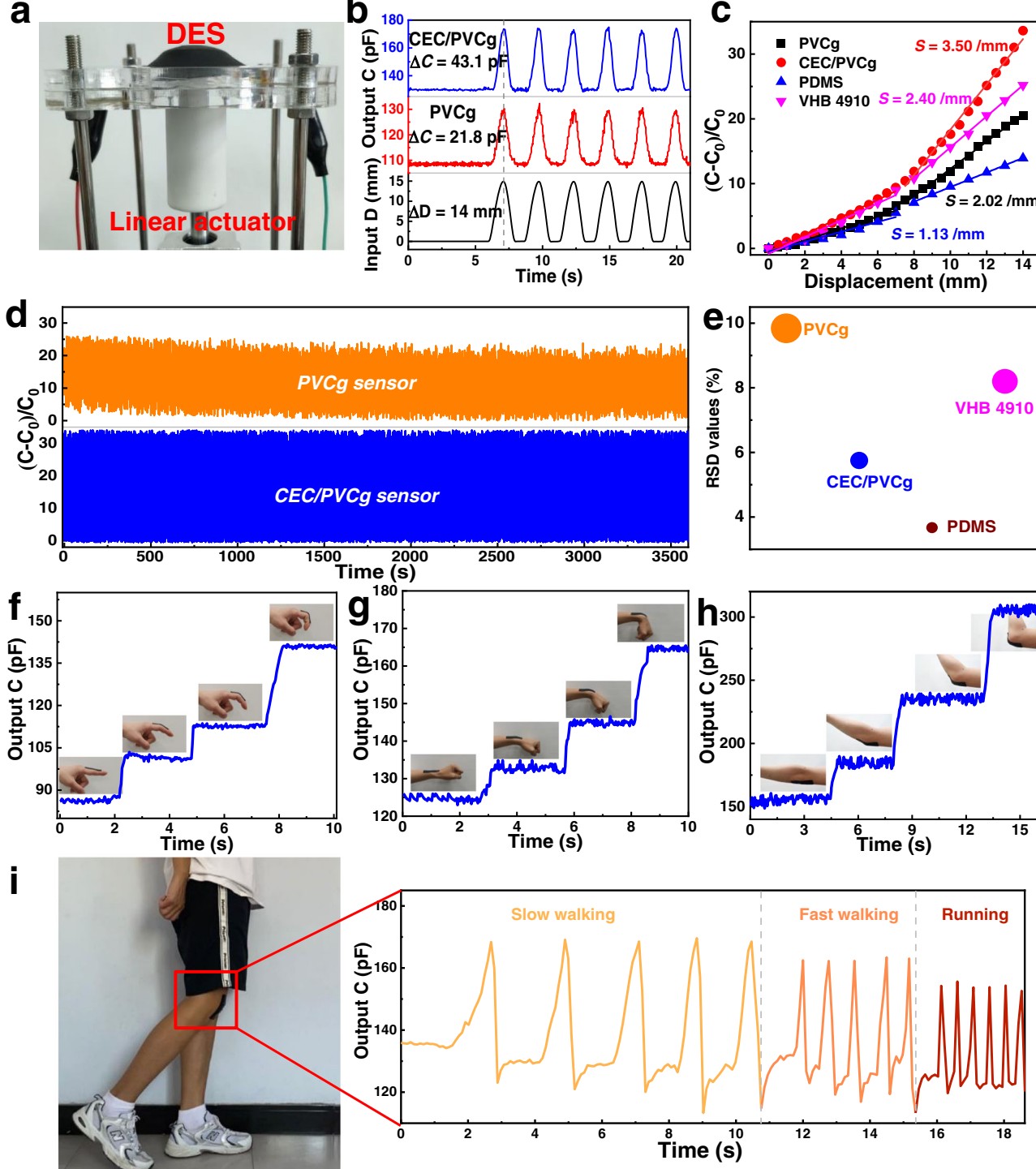

**Fig. 6 | Sensing test. a** A photograph of the DES device consisting of a linear actuator to apply periodic strains to the PVCg and CEC/PVCg-based sensors. **b** The input strain curve from linear actuator and output capacitance curves generated by the PVCg and CEC/PVCg sensors. **c** The relative capacitance ($C$-$C_0$/$C_0$) values that were generated by the PVCg, CEC/PVCg, PDMS, and VHB 4910 sensors as a function of the strain. The tangential slope of the curve was defined as the sensitivity (/mm) of the sensors. The values of sensitivity (S) in the displacement range of 7–14 were calculated and marked in the figure. **d** Duration test within 1440 cycle flections ($T$ = 2.5 s per cycle and 60 min in total) for PVCg and CEC/PVCg sensors. **e** Comparison of relative standard deviation (RSD = standard deviation of relative capacitance/mean of relative capacitance) for the four types of sensors. **f–h** Motions of human finger, wrist, and elbow are monitored by the CEC/PVC sensors. **i** Motions of human knee are monitored by the CEC/PVC sensors and transmitted into electrical signal online in real time with assistants of a wireless transmission module and a single-chip microcomputer.

the breakdown strength and further improve the actuation performances of DEAs. In addition, the evaluation of all the actuation and sensing performances of our devices were evaluated under room temperature. It has been reported that both breakdown strength and elastic modulus of DEs were sensitive to variations of the temperature depending on materials[53]. The temperature dependence needs to be evaluated if our DEA and DES are used at different environments in the future.

## Methods

### Materials
Polyvinyl chloride (PVC, $Mn = 47,000$) and microcrystalline cellulose (MC) were purchased from Adamas-Beta. Conductive silicone (ELASTOSIL® LR 3162 A/B) and silicone oil were obtained from Wacker Chemicals. Dioctyl phthalate (DOP), acrylonitrile (AN), and potassium thiocyanate (KSCN) were purchased from Sigma Aldrich.

### Synthesis and characterization of cyanoethyl cellulose (CEC)
CEC was synthesized by the following three steps: alkalization, addition, and purification. Alkalization: 2.5 g MC powder was added into 20 mL alkalizing water solution (containing 1.0 mol/L NaOH and 6.0 mol/L KSCN), and stirred for 30 min at room temperature, then the mixture was filtered to remove the remaining alkalizing solution, thus produced a white, OH-functionalized MC. Addition: the resultant OH-functionalized MC was dispersed and reacted with 20 mL AN solution at 30 °C for 3 h. Purification: The pH was adjusted to be 6.0 by using 1.0 mol/L acetic acid water solution, and then the mixture was filtered and washed three times by using deionized water and ethanol, until no thiocyanate ion was detected. Finally, CEC was dried in an oven at 75 °C for 24 h prior to use.

Transmittance FTIR (IFS66/S, Bruker, Germany) and XRD (D8 ADVANCE, Bruker, Germany) were used to monitor chemical component and crystalline structure. Elemental analyzer (VARIO ELIII, Elementar, Germany) was used to measure nitrogen contents of MC and CEC, for calculating the degree of substitution (DS) by using the equation (DS = $(162 \times N \%)/(1400-53 \times N \%)$).

### Preparation and characterization of CEC/PVCg elastomers
CEC/PVCg elastomers were prepared by solution casting method. CEC/PVCg elastomer with 9% CEC loading was prepared as following: 0.15 g CEC and 3.0 g DOP were added to 20.0 mL DMF solution, and magnetically stirred at 50 °C for 4 h until CEC completely dissolved into the solution. 1.5 g PVC powder was added to the mixture and continually stirred for 1 h at 50 °C. The mixture solution was poured into a self-made circle container, and then placed in a vacuum drying oven at 60 °C for 12 h, thus obtained a transparent CEC/PVCg elastomer film with CEC mass fraction ($fm$) of 9%, which was calculated by the equation ($fm = w_{CEC} / (w_{PVC} + w_{CEC})$). Similarly, CEC/PVCg elastomers with different loading concentrations of CEC (1.0–17 wt%) were obtained. The loading concentration of DOP, i.e., 2:1 mass ratio of PVC: DOP, was chose to balance the flexibility, viscoelastic effects, and breakdown strength of the resulting PVCg (Supplementary Note 1).

The former XRD, Raman (HR Evoulution, Horiba JY, France), and XPS (ESCALAB 250Xi, Thermo Scientific, USA) were used to test the chemical component of the PVC family films. Impedance analyzer (4294 A, Agilent, USA) was used to measure the dielectric permittivity, dielectric loss and conductivity of the PVC composites at room temperature in the frequency range of 40-$10^7$ Hz. A universal tensile testing machine (UTM2202, SUNS, China) was used to test their mechanical properties as well as the stress relaxation behaviors. Rotational rheometer (HAAKEMARS, Thermo Scientific, USA) was used to measure their storage modulus $G'$ and loss modulus $G''$ at room temperature.

### Assemblies and evaluations of DEA and DES
Following our previous report[16], CB/PDMS electrode was prepared and coated on both surfaces of the matrix films, thus produce a parallel capacitor for actuation and sensing. As controls, the commercial Dow Corning 186 and VHB 4910 acrylic elastomer were used to fabricate PDMS and VHB based DEA/DES, respectively. For DEA: A high voltage pulse power (ZMC, ZTE, China) was used to drive DEA, its duty ratio, driven voltage and frequency were respectively set to change in the ranges of 25–75%, 0.5–5.0 kV, and 0.5–100 Hz. The flection amplitude from the diaphragm center was collected by a laser displacement sensor (LK-G80A, Keyence, Japan) (Supplementary Fig. 11). The initial flection amplitude of 12 mm was generated by using the elastic spring as a pre-load, which corresponded to a pre-strain of 25%. An object-lifting test was used to evaluate DEA's recovery force. For DES: a linear motor was used to trigger the matrix to generate the periodic flections, its flection amplitude from the matrix center was collected by the laser displacement sensor for evaluating the DES effect (Supplementary Fig. 17). A commercial single-chip microcomputer (ATMEGA168P, WAYGAT, China) and a wireless transmission module (nRF24L01, GiSemi, China) were used to collect the capacitance signal.

## Data availability
The authors declare that the data that supports the findings of this manuscript can be found in the Supplementary Information and are available free of charge or available from the corresponding author upon request. Source data are provided with this paper.

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

## Acknowledgements

This work was supported by National Natural Science Foundation in China (52275295), and Central Plains Science and Technology Innovation Leading Talents (234200510026).

## Author contributions

G.D.J. and D.Y.H. contributed to the idea and designed the experiments. H.J.J. and Z.X.D. performed experiments and analyses, except polymer syntheses and characterizations, which was done by L.R.X. H.J.J. and D.Y.H. wrote the manuscript. All authors discussed the results and commented on the manuscript.

## Competing interests

The authors declare no competing interests.
