## [Peer Review File · Nature Communications]

Polyvinyl chloride-based dielectric elastomer with high permittivity and low viscoelasticity for actuation and sensingReviewers' Comments:

Reviewer #1:

Remarks to the Author:

In this manuscript, the authors produced a polyvinyl chloride (PVC)-based elastomer by the introduction of cyanoethyl cellulose (CEC) into a plasticized PVC. The PVC-based elastomer shows high permittivity, low viscoelasticity, but the dielectric loss is not satisfactory. Besides a few issues need to be clarify.

1. It is unclear how the plasticized PVC named by the authors is processable. It is confusing that the authors named the matrix as plasticized PVC (in line 92), pristine PVC (in line 94), PVC (in Fig.2e), and the author should provide an explicit concept. Besides, the mass fraction of DOP exceeds 50%, Why the authors provided no more explanation about the role of DOP in the performance of PVC-based elastomer?
2. Why the authors chose the electric field of 9.09 V/ μm for actuation test? The electric strength of all new DEs in this work needs to be supplied.
3. In line 69-71, the authors claimed that "DE actuators with low dielectric permittivity often require high electric field to drive ($> 20 \text{ V}/\mu\text{m}$), leading to the high risks of current leakage and electrical breakdown", but it is inaccurate. If the electric field to drive is far lower than the breakdown strength while just a large value, the current leakage and electrical breakdown may not happen.
4. According to the Supplementary Fig.7, the actuation test of DEs may be under prestrain condition, and the authors need to clarify this point. About the results of actuation test, the authors cited some references to highlight the driving properties of DEs in this work, while the results of actuation test in the references of 9, 22, 35, were obtained under non-prestrain condition. Besides, the work of reference 36, is about polyurethane dielectric elastomer, not about VHB.
5. About stress relaxations of DEs, in line 195 the constant strain is 200%, while in line 205 the constant strain is 100%?
6. Except of Fig 3, the quality of other Figures needs to be improved.
7. Some crucial references should be cited in order to make a good understanding to this work. They include Chen et al. Nature 2019, 575, 324-329; Chen et al. Chemical Engineering Journal 2021, 405, 126634; Cao et al. Extreme Mechanics Letters 2020, 35, 100619; Feng et al. Chemical Reviews 2022, 122, 3820-3878 and Yin et al. Nature Communications 2021, 12, 4517.

Reviewer #2:

Remarks to the Author:

Background: Dielectric elastomers show great promise as large strain artificial muscle. A challenge has been the high voltages and field strengths needed, with the devices operating close to breakdown and requiring kV level sources. If the voltage levels can be reduced to about 1 kV or less, the electronics is much less expensive. In order to achieve lower voltage operation, many investigators have sought to increase dielectric constant. This generally does not lead to improved strain, as voltage drops but so does breakdown strength, and elastic modulus rises. Another challenge is that the most commonly used elastomer - VHB - is highly viscoelastic in its behaviour, and so there is loss and creep. This is not the case in some silicones, which can achieve high bandwidth, relatively low loss, actuation - though silicones still requires high field strength. Kornbluh has achieved kHz frequency actuation of silicone, showing low loss.

The authors show that plasticized PVC with a high dielectric constant additive generate significant area strain (12%) at low field strength (9 kV/mm) compared to VHB, regular PVC, silicone and others. This is still a field strength well above the breakdown strength of air, but it is significantly lower than commonly used, and so is an important advance. The PVC additive used increases modulus, but produces a larger increase in dielectric constant, and reduces viscoelastic loss. Overall this leads to good low voltage actuation.

The authors compare creep and stress relaxation to that in the literature, for example in Figure 5d. How can a fair comparison be made, given the different loading conditions?

The authors have not expressed load in terms of stress, or described in the main text the amount of pre-strain used.

There has been substantial work on lowering voltage in dielectric elastomers by increasing dielectric constant, or reducing thickness. Early work on dielectrics was by Kofod I believe, and there has been a lot since, while a number of groups have sought to make thin elastomer layers. This work has not been properly summarized by the authors.

The improved sensor response compared to PVC is well presented. It is unclear how this performance compares to that of VHB and silicones.

What is the dielectric loss at typical operating frequencies/timescales - e.g. 1 Hz?

It is claimed that the actuators and sensors show fast response. I don't see a frequency response or time response analysis, or comparison with other materials.

What are the potential drawbacks of the approach? Do we expect a greater temperature dependence? Is breakdown strength reduced? Can large actuation be achieved, by applying higher voltages (why did the authors stop at the 9 V/micron)?

The sensor work is well presented. It is important to put this work in context of other capacitive sensor work, and perhaps provide some detail in the supplementary. There are also commercial sensors (e.g. Stretchsense) to compare with.

Overall, the paper makes a significant contribution to the field of electroactive polymers. It is written clearly and concisely. I would recommend considering it for publication, following revisions.

Reviewer #3:

Remarks to the Author:

The authors proposed a polyvinyl chloride (PVC) -based dielectric elastomer with high permittivity and low viscoelasticity by introduction of cyanoethyl cellulose (CEC) into a plasticized PVC. The physicochemical and electromechanical coupling property of the as-synthesized CEC/PVC were carefully studied. This work improved the performance of traditional PVC. However, the demonstrated performance of actuation and sensing of the presented material are not convincing to me. The paper cannot meet the standard of Nature Communications unless the authors could clarify the advantage of the material compared to the existing material.

The strategy of increasing permittivity and reducing viscosity in the material synthesis part sounds reasonable and interesting. However, if the material is aimed to serve for dielectric elastomer applications, these two indexes are apparently not dominated material properties. As shown in Fig. 5, the actuation strain of the new material falls within the order of 10%. A lot of existing studies have shown that various of materials without careful optimization and complicated synthesis can easily achieve this actuation level. Similar concerns apply for the sensing demo in this work. In the introduction, the authors pointed out that the VHB material, widely used in literature, has a clear drawback in viscous property. However, the synthesized new material improves its viscous property compared to VHB but significantly sacrifices its ability of large actuation, which, to the reviewer's opinion, is not satisfactory for dielectric elastomer applications.

**Responses to Reviewer #1:**

We thank the reviewer for her/his insightful report. We are delighted that the reviewer agreed that “*The PVC-*
*based elastomer shows high permittivity, low viscoelasticity*”. Here we address the comments and technical
questions raised by the reviewer with new experimental results and analyses.

**COMMENT 1#:** *The PVC-based elastomer shows high permittivity, low viscoelasticity, but the dielectric loss*
*is not satisfactory.*

**Response:** The heat generated due to dielectric losses of dielectric elastomers (DEs) would result in substantial
increases in their temperature and conductivity over time, which would lower the breakdown strength and may
lead to thermal or electrical breakdown [R1-R3]. Therefore, low levels of dielectric loss < 0.3 are highly
preferred for DEA application as suggested by previous study [R4,R5]. However, the commonly used
strategies for increasing dielectric permittivity, such as the addition of inorganic polar particles and conductive
particles, are often associated with substantial increases in dielectric loss (e.g. > 0.5), leading to current leaking
and/or electric breakdown. For instance, the addition of multiwalled carbon nanotube with 6-9 wt% loadings
into PDMS precursors resulted in very high dielectric losses of 3.74-4.00 @ 1.0 kHz [R6]. By contrast, the
dielectric loss of CEC/PVCg elastomers were maintained in a low value range 0.033-0.142 in this study (**Fig.**
**R1a**). To address the reviewer’s concern, we measured the conductivity and breakdown strength of CEC/PVCg
elastomers. Because according to dielectric percolation theory, the increase of dielectric loss is resulted from
the formation of a connected conductive network by the conductive filler materials. The results showed that
CEC/PVCg elastomers with conductivity range of 3.40×10^{-9} - 4.55×10^{-9} S/cm were in a highly insulating state
(**Fig. R1b** and **R1c**). Moreover, it was found that the addition of CEC only slightly decreased the breakdown
strength. The breakdown strength of CEC/PVCg elastomers were 18.22-20.06 V/ μ m (**Fig. R1a**). Although a
low electric field of 9.09 V/ μ m is applied to drive the DEA in this study, a large actuation strain, which is
much larger than commercial VHB 4910 based DEA has been achieved with our CEC/PVCg DEA. Therefore,
we believe the dielectric loss of CEC/PVCg elastomers here is low enough to support the large actuation while
preventing the device breakdown.

**Fig. R1** Dielectric property tests. Dielectric losses and breakdown strengths of CEC/PVCg elastomers with
CEC mass loadings of 0-17 wt% (a). AC conductivities of CEC/PVCg elastomers with varying CEC loadings
under 40 - 10^7 Hz frequencies (b). Evolution of AC conductivity at 1 kHz for the CEC/PVCg elastomers (c).

The data has now been added into the revised manuscript as **Fig. 3c** and the relevant discussion at
Page 7, Paragraph 3, as shown below:

“Dielectric loss is crucial for most DEs. Because the high levels of dielectric loss can result in substantial
increases in both temperature and conductivity, which could potentially lead to thermal or electrical

breakdown⁴³⁻⁴⁵. Unfortunately, the commonly used strategies of increasing dielectric permittivity, such as the
 addition of inorganic polar particles and conductive particles, were often associated with a substantial increase
 in the dielectric loss (e.g. > 0.5 at 1 kHz), which would lower the breakdown strength and reduce the lifetime
 of actuators^{46,47}. According to dielectric percolation theory, the increase of dielectric loss is mainly due to the
 increase of conductive filler materials and the formation of a connected conductive network. For instance, the
 addition of 6-9 wt% multiwalled carbon nanotube into PDMS resulted in very high dielectric losses of 3.74-
 4.00 @ 1.0 kHz⁴⁸. By contrast, the introduction of CEC into PVCg matrix in this study induced a marginal
 increase, resulting in dielectric losses of < 0.15 at 1 kHz for CEC/PVCg elastomers (**Fig. 3c**), which are still
 in a low level for DEs as suggested by previous study⁴⁹. The limited increase of dielectric loss following the
 addition of CEC could be ascribed to the highly electrical insulating nature of CEC (**Supplementary Note 2**).
 The measurement of conductivity indicated that the CEC/PVCg elastomers (4.0×10^{-9} S/cm) were in a highly
 insulating state. Importantly, the breakdown strength was only slightly decreased by 10.4 % following the
 addition of CEC and kept almost constant for CEC/PVCg elastomers with different loading concentrations of
 CEC (**Fig. 3c** and **Supplementary Information Fig. S4**). The detected breakdown strength of CEC/PVCg
 elastomers with 9 wt% CEC was 19.53 V/ μ m. Thus, a much lower driving electric field of 9.09 V/ μ m was
 used in all of experiments in this study in order to prolong the working life of device and achieve a stable
 performance while generating a desired large area strain.”

 **COMMENT 2#:** *It is unclear how the plasticized PVC named by the authors is processable. It is confusing*
 *that the authors named the matrix as plasticized PVC (in line 92), pristine PVC (in line 94), PVC (in Fig.2e),*
 *and the author should provide an explicit concept. Besides, the mass fraction of DOP exceeds 50%, Why the*
 *authors provided no more explanation about the role of DOP in the performance of PVC-based elastomer?*

**Response:** We thank the reviewer for her/his suggestion. We have corrected the manuscript by naming pure
 PVC as pure PVC and naming plasticized PVC as PVC gel (PVCg) because of the plasticized PVC is in a gel-
 like form.

The major purpose of adding a plasticizer, *i.e.* DOP in this study, into PVC matrix is to increase the
 compliance and flexibility of PVC, which is critical for their actuation and sensing performances. As per the
 actuation strain equation of $S_z = \epsilon_r \epsilon_0 E^2 / Y$, the matrix film with a lower modulus (Y) would generate a larger
 actuation strain (S) [R7-R9]. It has been suggested that DEs with a modulus less than 1 MPa is preferred for
 actuation application [R7]. The pure PVC has a very high modulus of 23.2 MPa (**Fig. R2a** and **R2c**), which is
 not suitable for actuation and sensing applications. The introduction of DOP would weaken the molecular
 interactions among PVC chains, leading to the transition of PVC matrix from glassy-state to hyperelastic-state
 with the high flexibility [R10]. Our results showed that the addition of 50-80 wt% DOP into PVC matrix
 significantly reduced the moduli to 0.02-0.54 MPa and increased the elongation at break to 200-560 % (**Fig.**
 **R2b** and **R2c**), making the matrix more favorable for actuation and sensing applications.

**Fig. R2** Mechanical property tests. Stress-strain curves of the self-casting (a) PVC plastics and (b) PVCg
 elastomers with PVC: DOP mass ratios of 1:1, 1:2, 1:3, 1:4, and (c) their Young's moduli.

Although the introduction of plasticizer is a common strategy to generate large actuation of PVC-
 based DEs, the plasticized PVC suffers from inherent strong viscoelastic effects, which results in evident
 mechanical loss, stress relaxation, and viscoelastic hysteresis, eventually leading to instability and attenuation
 of output signals over time as well as delayed response in actuation and sensing applications [R11, R12]. Both
 dynamic and static viscoelasticity of plasticized PVC, *i.e.* PVCg, as a function of the loading concentrations
 of DOP were measured and shown in the following **Fig. R3** and **Fig. R4**, respectively. The dynamic
 viscoelasticity, *i.e.* mechanical loss of PVCg ($\tan \delta = G''/G'$), was increased by 15.50 folds by increasing the
 loading concentration of DOP from 50 wt% (1:1) to 80 wt% (1:4) (**Fig. R3**). Under constant load of 60 g for
 6 hours, the creep-induced elongation increased from 25 % for PVCg (1:1) to 60 % for PVCg (1:4) (**Fig. R4a-
 4d**). Similarly, under constant strain of 100 % for 10 min, the recorded stress attenuation increased from 11.41 %
 for PVCg (1:1) to 33.99 % for PVCg (1:4) (**Fig. R4e and 4f**). Notably, currently existing PVC matrix used as
 DEA or DES often contain much larger concentrations of plasticizer (> 90 wt%) than this study [R13], leading
 to even stronger viscoelastic effects than what we demonstrated here. In addition, the introduction of DOP
 decreased the breakdown strength of PVC matrix due to the percolation effect of DOP, as shown in the
 following **Fig. R5a**.

To conclude, the increase of plasticizer content in PVCg resulted in lower elastic modulus (*i.e.* higher
 flexibility), higher viscoelastic effects, and a lower breakdown strength. Therefore, we choose the mass ratio
 of 1: 2 (PVC: plasticizer) to balance these properties of PVCg. More importantly, the introduction of CEC into
 the plasticized PVCg can address this long-standing challenge by significantly reducing its viscoelastic effects
 and concurrently achieving the high permittivity.

 **Fig. R3** Dynamic viscoelasticity tests of PVCg elastomers with varying DOP concentrations. Evolutions of (a)
 storage moduli (G'), (b) loss moduli (G''), and (c) the counted mechanical losses ($\tan \delta = G''/G'$) under
 frequencies of 0.01-10 Hz.

**Fig. R4** Static viscoelasticity of PVCg elastomers with varying DOP concentrations. Creep behaviors for PVCg
 elastomers with PVC: DOP mass ratios of (a) 1:1, (b) 1:3, (c) 1:4 under a constant load of 60 g for 6 hours
 (data of the elastomer of mass ratio of 1:2 was shown formerly). Quantifications of (d) creep strains, (e) stress
 relaxations, and (f) percentages of stress attenuation of PVCg elastomers with PVC: DOP mass ratios of 1:1,
 1:2, 1:3, 1:4.

**Fig. R5** Measurement of breakdown strengths. (a) Evolution of breakdown strengths for PVCg elastomers
 with varying DOP concentrations. (b) Evolution of breakdown strengths for CEC/PVCg elastomers with
 varying CEC concentrations.

We have now added the new data into the revised supplementary materials and relevant discussion
 about the rationale and impacts of the introduction of DOP into PVC in the **Supplementary Note 1** and the
 revised manuscript at Page 4, Paragraph 2 and Page 19 and Paragraph 2, as shown below:

“The plasticizers are often introduced into PVC matrices in order to produce highly flexible PVCg elastomers
 with high flexibility by weakening the interaction forces among PVC chains²⁹ (**Supplementary Information**
 **Fig. S1**). However, the plasticized PVCg suffer from low breakdown strength and inherent strong viscoelastic
 effects³⁰, which leads to time-dependent change of internal stress and strain³¹, *i.e.* creep (**Supplementary**
 **Information Fig. S3**).”

“The loading concentration of DOP, *i.e.* 2:1 mass ratio of PVC: DOP, was chose to balance the flexibility,

viscoelastic effects, and breakdown strength of the resulting PVCg (Supplementary Note 1).”

*COMMENT 3#:* Why the authors chose the electric field of 9.09 V/μm for actuation test? The electric strength
of all new DEs in this work needs to be supplied.

**Response:** Existing DEAs often requires high driving electrical field to achieve large strain, which, however,
could potentially lead to polymer creep, current leakage, and electrical breakdown. In addition, it also needs a
bulky, high-voltage power supply system, hampering its wide-spread applications [R14]. Therefore, generation
of a large strain under low driving voltages is highly desirable for DEAs, which remains a challenge. In this
study, we have measured the electric breakdown strength of all DEs, as shown in the Fig. R5. The breakdown
strength of plasticized PVCg (1:2) was about 20 V/μm. The introduction of CEC with various loading
concentrations have limited impacts on the breakdown strength while it dramatically increased permittivity.
Thus, a relative low working voltage of 9.09 V/μm was selected in order to prolong the working life of the
device, achieve a stable performance while generating a desired large area strain [R15-18]. Notably, such strain
of 12.22 % was achieved under a very small pre-strain of 25 %, which was negligible when compared to other
pre-strains such as 540 % and 400 % [R19, R20]. We have now provided the new data of breakdown strength
measurement in the revised manuscript as Fig. 3c and discussions on the reason for choosing the low electric
field of 9.09 V/μm at Page 8, Paragraph 1, as shown below:

“Importantly, the breakdown strength was only slightly decreased by 10.4 % following the addition of CEC
and kept almost constant for CEC/PVCg elastomers with different loading concentrations of CEC (Fig. 3c and
Supplementary Information Fig. S4). The detected breakdown strength of CEC/PVCg elastomers with 9 wt%
CEC was 19.53 V/μm. Thus, a much lower driving electric field of 9.09 V/μm was used in all of experiments
in this study in order to prolong the working life of device and achieve a stable performance while generating
a desired large area strain.”

*COMMENT 4#:* In line 69-71, the authors claimed that “DE actuators with low dielectric permittivity often
require high electric field to drive (> 20 V/μm), leading to the high risks of current leakage and electrical
breakdown”, but it is inaccurate. If the electric field to drive is far lower than the breakdown strength while
just a large value, the current leakage and electrical breakdown may not happen.

**Response:** We agree with the reviewer that when the driving electric field is far lower than the breakdown
strength, the current leakage and electrical breakdown should not happen. Unfortunately, the existing DE
actuators often required the high driving electric fields that are close to their breakdown strength in order to
achieve large strain because of their low permittivity. For example, as we measured in this study, VHB-based
actuators produced rather small area strain of 3.45 % under driving electric field of 22.5 V/μm, which is close
to their breakdown strength of 28.4 V/μm [R15] (Supplementary Table S2). PDMS (Gelest OE™ Extended
Cure)-based actuator required 30 V/μm, which was the measured breakdown strength, to achieve the area strain
of 4.63 % [R21]. In addition, even the driving voltage is far lower than breakdown strength, a high value of
several kilovolts arises safety issues and brings a problem of using a bulky high-voltage power supply system.
To address the reviewer’s concern, we have updated our description and added more discussions on this issue
in the revised manuscript Page 3 and Paragraph 1, as shown below:

“DE actuators with low dielectric permittivity, such as PDMS and VHB materials, often require high driving
electric fields (> 20 V/μm) to achieve large actuations, which would lead to the high risks of current leakage¹⁵
and electrical breakdown¹⁶ when such high driving electric fields are close to their breakdown strength. In
addition, a high value of several kilovolts arises safety issues and brings about the problem of using a bulky
high-voltage power supply system¹⁷.”

**COMMENT 5#:** According to the Supplementary Fig.7, the actuation test of DEs may be under prestrain
condition, and the authors need to clarify this point. About the results of actuation test, the authors cited some
references to highlight the driving properties of DEs in this work, while the results of actuation test in the
references of 9, 22, 35, were obtained under non-prestrain condition. Besides, the work of reference 36, is
about polyurethane dielectric elastomer, not about VHB.

**Response:** We thank the reviewer for bringing up this important issue. We have now clarified that the DEAs
were under 25 % pre-strain in this study in the revised manuscript. We have corrected our citation of Reference
36. Moreover, we have prepared the PDMS and VHB 4910-based DEAs and measured their actuation
performances under the exact same conditions in order to make a fair comparison to our CEC/PVCg-based
actuators. As shown in the following **Fig. R6**, the detected area strains were 1.46-12.22 % for the CEC/PVCg
actuators under the driving electrical fields of 5.45-9.09 V/ μm , 0.81-3.15 % for the PVCg actuators under 5.45-
9.09 V/ μm , 0.65-2.44 % for the PDMS actuators under 5.85-9.07 V/ μm , while VHB 4910 actuators cannot be
activated under the driving electrical field < 12.5 V/ μm , *i.e.* strain of 0 %. The area strains of VHB 4910
actuators were 0.72-3.45 % when the driving electrical fields were further increased to 12.5-22.5 V/ μm . To
conclude, our CEC/PVCg actuators produced the largest actuation strains under low driving voltages as
compared to the commonly used PDMS and VHB-based actuators. We have now provided the new data as
**Fig. 6b** and relevant discussions in the revised manuscript at Page 11, Paragraph 3, and Page 20, Paragraph 1,
as shown below:

“According to the strain model shown in **Supplementary Information Fig. S12**, the counted area strain
generated by the CEC/PVCg actuators was 12.22 % (with 9 wt% CEC, under driving voltage of 9.09 V/ μm ,
pre-strain of 25 %), which represents 3.9-fold increase as compared to the PVCg actuators (**Fig. 5b**). In addition,
we prepared the commonly used PDMS and VHB-based actuators and measured their actuation strains under
the same conditions as the comparison (**Fig. 5b, Supplementary Table S2, and Supplementary Information**
**Fig. S15**). PDMS actuators produced strains of 0.65-2.44 % under electrical fields of 5.85-9.07 V/ μm . VHB
4910 actuators could not be triggered, *i.e.* 0 % area strain, under electrical fields < 12.5 V/ μm and produced
only 0.72-3.45 % strains by further increasing electrical fields to 12.5-22.5 V/ μm . Therefore, our CEC/PVCg
actuators generated significantly larger actuation strains, *i.e.* > 5 times, than commonly used PDMS and VHB
4910 actuators, which was largely attributed to the augmentation of the electromechanical coupling sensitivity
k of CEC/PVCg (**Fig. 3e**).”

“The initial flexion amplitude of 12 mm was generated by using the elastic spring as a pre-load, which
corresponded to a pre-strain of 25 %.”

**Fig. R6** Evaluation of actuation properties of (a) PVCg, (b) CEC/PVCg, (c) PDMS, (d) VHB 4910 actuators
by measuring their flexion displacements and area strains under various driving voltages. (e) The comparison
of area strains (mean values) that were generated by different actuators.

*COMMENT 6#:* About stress relaxations of DEs, in line 195 the constant strain is 200 %, while in line 205
the constant strain is 100 %?

**Response:** 200 % was a typo and we have corrected the value of constant strain to 100 % in the revised
manuscript.

*COMMENT 7#:* Except of Fig 3, the quality of other Figures needs to be improved.

**Response:** We have now improved the quality and resolution of all the figures. We have now submitted all
figures in the format of .TIFF instead of word file in the initial submission, which reduce the resolution of
figures.

*COMMENT 8#:* Some crucial references should be cited in order to make a good understanding to this work.
They include Chen et al. *Nature* 2019, 575, 324-329; Chen et al. *Chemical Engineering Journal* 2021, 405,
126634; Cao et al. *Extreme Mechanics Letters* 2020, 35, 100619; Feng et al. *Chemical Reviews* 2022, 122,
3820-3878 and Yin et al. *Nature Communications* 2021, 12, 4517.

**Response:** We thank the reviewer for bringing these important articles to our attention. We have now added
and discussed the following references in the revised manuscript as shown below:

“DE actuators (DEAs) are attractive artificial muscles due to their high energy density⁷ and conversion
efficiency⁸, and fast response⁹.”

“For example, the dielectric permittivity of widely used DEs are 2.2-3.0 for polydimethylsiloxane (PDMS)¹¹,
4.4-4.7 for VHB acrylic elastomer (3M)^{12,13}, and 4.0 for pure polyvinyl chloride (PVC)¹⁴.”

“In addition, a high value of several kilovolts arises safety issues and brings about the problem of using a
bulky high-voltage power supply system¹⁷.”

7. Chen, Y. F. et al. Controlled flight of a microrobot powered by soft artificial muscles. *Nature* **575**, 324-
329 (2019).

8. Feng, Q. K. et al. Recent progress and future prospects on all-organic polymer dielectrics for energy
storage capacitors. *Chem. Rev.* **122**, 3820-3878 (2022).

9. Cao, C. J., Gao, X., Burgess, S. & Conn, A. T. Power optimization of a conical dielectric elastomer
actuator for resonant robotic systems. *Extreme Mech. Lett.* **35**, 100619 (2020).

12. Yin, L. J. et al. Soft, tough, and fast polyacrylate dielectric elastomer for non-magnetic motor. *Nat.*
*Commun.* **12**, 4517 (2021).

17. Chen, Z. Q. et al. Ultrasoft-yet-strong pentablock copolymer as dielectric elastomer highly responsive to
low voltages. *Chem. Eng. J.* **405**, 126634 (2021).

References

- [R1] Khanchaitit, P., Han, K., Gadinski, M. R., Li, Q. & Wang, Q. Ferroelectric polymer networks with high energy density and improved discharged efficiency for dielectric energy storage. *Nat. Commun.* **4**, 2845 (2013).
- [R2] Madsen, F. B., Yu, L., Mazurek, P. S. & Skov, A. L. A simple method for reducing inevitable dielectric loss in high-permittivity dielectric elastomers. *Smart. Mater. Struct.* **25**, 075018 (2016).
- [R3] Shamsul, Z., Morshuis, P. H. F., Yahia, B. M., Gernaey, K. V. & Skov, A. L. The electrical breakdown of thin dielectric elastomers: thermal effects. *Proc. SPIE* **9056**, 90562V (2014).
- [R4] Opris, D. M. et al. New silicone composites for dielectric elastomer actuator applications in competition with acrylic foil. *Adv. Funct. Mater.* **21**, 3531-3539 (2011).
- [R5] Guan, J. P., Xing, C. Y., Wang, Y. Y., Li, Y. J. & Li, J. Y. Poly (vinylidene fluoride) dielectric composites with both ionic nanoclusters and well dispersed graphene oxide. *Compos. Sci. Technol.* **138**, 98-105 (2017).
- [R6] Kohlmeyer, R. R. et al. Electrical and dielectric properties of hydroxylated carbon nanotube-elastomer composites. *J. Phys. Chem. C* **113**, 17626-17629 (2009).
- [R7] Madsen, F. B., Daugaard, A. E., Hvilsted, S. & Skov, A. L. The current state of silicone-based dielectric elastomer transducers. *Macromol. Rapid Commun.* **37**, 378-413 (2016).
- [R8] Romasanta, L. J., Lopez-Manchado, M. A. & Verdejo, R. Increasing the performance of dielectric elastomer actuators: a review from the materials perspective, *Prog. Polym. Sci.* **51**, 188-211 (2015).
- [R9] Plante, J. S. & Dubowsky, S. Large-scale failure modes of dielectric elastomer actuators. *Int. J. Solids Struct.* **43**, 7727-7751 (2006).
- [R10] Daniels, P. H. A brief overview of theories of PVC plasticization and methods used to evaluate PVC-plasticizer interaction. *J. Vinyl. Addit. Technol.* **15**, 219-223 (2009).
- [R11] Tan, M. W. M., Thangavel, G. & Lee, P. S. Enhancing dynamic actuation performance of dielectric elastomer actuators by tuning viscoelastic effects with polar crosslinking. *NPG Asia Mater.* **11**, 62 (2019).
- [R12] Zou, J. & Gu, G. Y. Modeling the viscoelastic hysteresis of dielectric elastomer actuators with a modified rate-dependent prandtl-ishlinskii model. *Polymers* **10**, 525 (2018).
- [R13] Li, Y. & Hashimoto, M. PVC gel based artificial muscles: Characterizations and actuation modular constructions. *Sens. Actuators A: Phys.* **233**, 246-258 (2015).
- [R14] Chen, Z. Q. et al. Ultrasoft-yet-strong pentablock copolymer as dielectric elastomer highly responsive to low voltages. *Chemical Engineering Journal. Chem. Eng. J.* **405**, 126634 (2021).
- [R15] Yin, L. J. et al. Soft, tough, and fast polyacrylate dielectric elastomer for non-magnetic motor. *Nat. Commun.* **12**, 4517 (2021).
- [R16] Shankar, R., Ghosh, T. K. & Spontak, R. J. Electromechanical response of nanostructured polymer systems with no mechanical pre-strain. *Macromol. Rapid Commun.* **28**, 1142-1147 (2007).
- [R17] Zhao, Y. et al. Remarkable electrically actuation performance in advanced acrylic-based dielectric elastomers without pre-strain at very low driving electric field. *Polymer* **137**, 269-275 (2018).
- [R18] Yang, C. X., Gao, X. & Luo, Y. W. End-block-curing ABA triblock copolymer towards dielectric elastomers with both high electro-mechanical performance and excellent mechanical properties. *Chem. Eng. J.* **382**, 123037 (2020).
- [R19] Pelrine, R., Kornbluh, R., Pei, Q. B. & Joseph, J. High-speed electrically actuated elastomers with strain greater than 100%. *Science* **287**, 836-845 (2000).
- [R20] Huang, J. S. et al. Giant, voltage-actuated deformation of a dielectric elastomer under dead load. *Appl. Phys. Lett.* **100**, 041911 (2012).
- [R21] Sadroddini, M. & Kashani, M. R. Silica-decorated reduced graphene oxide (SiO₂@rGO) as hybrid fillers

for enhanced dielectric and actuation behavior of polydimethylsiloxane composites. *Smart Mater. Struct.*
**29**, 015028 (2020).

**Responses to Reviewer #2:**

We are grateful for reviewer's detailed and positive comments on our manuscript. We thank the reviewer for
his/her very encouraging remarks that "Overall, the paper makes a significant contribution to the field of
electroactive polymers. It is written clearly and concisely. I would recommend considering it for publication,
following revisions." We are also delighted that the reviewer agreed that the viscoelastic effects of our device
is "significantly lower than commonly used, and so is an important advance", "The PVC additive used
increases modulus, but produces a larger increase in dielectric constant, and reduces viscoelastic loss. Overall,
this leads to good low voltage actuation" Here we address the comments and the technical questions from the
reviewer with the new experimental data and analysis.

**COMMENT #1:** *The authors compare creep and stress relaxation to that in the literature, for example in*
*Figure 5d. How can a fair comparison be made, given the different loading conditions?*

**Response:** To address reviewer's concern, we have prepared PDMS and VHB 4910-based actuators and
evaluated their actuation performances under the exact same conditions as our CEC/PVCg actuators. First, the
actuation displacements over 1000 cycles, *i.e.* 1000 seconds of four types of actuators were measured and the
relative displacement shifts (RDS) were calculated to quantify their viscoelastic drifts. As shown in the
following **Fig. R1a-1e**, the RDS values of VHB 4910, PDMS, PVCg, and CEC/PVCg actuators were 136.09 %,
5.70 %, 59.40 %, and 7.78 %, respectively. Our CEC/PVCg actuators showed a very low shift of displacement
over 1000 cycles of actuation, which was 94 % and 87 % reductions as compared to VHB 4910 and PVCg
actuators. Second, the area strains of four types of DEAs were measured (**Fig. R1f**). The detected area strains
of PDMS, PVCg, and CEC/PVCg actuators were 2.44 %, 3.15%, and 12.22 %, respectively under the driving
electric field of 9.09 V/ μm . VHB 4910 actuators cannot be activated under the driving electrical field < 12.5
V/ μm , *i.e.* strain of 0 %, and showed area strains of 0.72 %-3.45 % when the driving electrical fields were
further increased to 12.5-22.5 V/ μm . Our CEC/PVCg actuators showed > 4-fold increases in area strains as
compared to other three types of actuators. To conclude, Our CEC/PVCg actuators showed low viscoelastic
effects and large area strains, demonstrating the significant improvement in actuation performances as
compared to the existing dielectric elastomer actuators, such as PDMS and VHB 4910.

We have now provided the updated data of area strains and displacement shifts as **Fig. 5b** and **5d** in
the revised manuscript. The relevant descriptions and discussions were provided in the revised manuscript at
Page 11, Paragraph 3 and Page 12, Paragraph 2, as shown below:

"According to the strain model shown in **Supplementary Information Fig. S12**, the counted area strain
generated by the CEC/PVCg actuators was 12.22 % (with 9 wt% CEC, under driving voltage of 9.09 V/ μm ,
pre-strain of 25 %), which represents 3.9-fold increase as compared to the PVCg actuators (**Fig. 5b**). In addition,
we prepared the commonly used PDMS and VHB-based actuators and measured their actuation strains under
the same conditions as the comparison (**Fig. 5b, Supplementary Table S2, and Supplementary Information**
**Fig. S15**). PDMS actuators produced strains of 0.65-2.44 % under electrical fields of 5.85-9.07 V/ μm . VHB
4910 actuators could not be triggered, *i.e.* 0 % area strain, under electrical fields < 12.5 V/ μm and produced
only 0.72-3.45 % strains by further increasing electrical fields to 12.5-22.5 V/ μm . Therefore, our CEC/PVCg
actuators generated significantly larger actuation strains, *i.e.* > 5 times, than commonly used PDMS and VHB
4910 actuators, which was largely attributed to the augmentation of the electromechanical coupling sensitivity
*k* of CEC/PVCg (**Fig. 3e**)."

"The displacements of the PVCg, CEC/PVCg, PDMS, and VHB 4910 actuators were measured and recorded
over actuation for 1000 cycles, *i.e.* 1000 seconds (**Fig. 5c** and **Supplementary Information Fig. S16**). The
CEC/PVCg and PDMS actuators produced remarkably stable displacement profiles. By contrast, apparent

displacement shifts were observed over time for PVCg and VHB 4910 actuators. Relative displacement shifts
 (RDS) were calculated to quantify the viscoelastic effects (Fig. 5d). It was found that RDS values increased
 with time, *i.e.* number of actuation cycles, for VHB 4910 and PVCg actuators, while remaining almost constant
 for CEC/PVCg actuators. The relative shifts over 1000 cycles of CEC/PVCg actuators (7.78 % of RDS)
 represented 87 % and 94 % reductions as compared to PVCg (59.40 % of RDS) and VHB 4910 actuators
 (136.09 % of RDS). PDMS actuators (5.70 % of RDS) displayed similar viscoelastic drifts to CEC/PVCg
 actuators.”

 **Fig. R1** Actuation stability and area strains of four types of actuators. Duration tests for the (a) PVCg, (b)
 CEC/PVCg, (c) VHB 4910, and (d) PDMS actuators over 1000 cycles of actuation, *i.e.* 1000 seconds. (e)
 Evolutions of relative displacement shift (RDS) of the four types of actuators. The actuation tests of PVCg,
 CEC/PVCg, and PDMS were performed under 9.09 V/ μ m electrical field and 1 Hz frequency, while the VHB
 actuator was triggered under 22.5 V/ μ m electrical field and 1 Hz frequency. $RDS = \frac{|D - D_{creep}|}{D} \times 100\%$, where
 D is the amplitude of displacement and D_{creep} is the shift of the displacement as shown in (a). (f) The mean
 values of area strains that were generated by four types of actuators as functions of driving electric fields.

**COMMENT #2:** The authors have not expressed load in terms of stress, or described in the main text the
 amount of pre-strain used.

**Response:** In the evaluation of actuation performance, the elastomer film was carefully coated on a circle
 frame ($\phi = 50$ mm) to make a concise and facile pump diaphragm. By using an elastic spring as the normal

preload, the diaphragm center was pulled down 12 mm each time and the counted area pre-strain was 25 %
according to the spherical crown model shown in **Supplementary Information Fig. S12**. We have now
provided the description of pre-strain in the revised manuscript at Page 20, Paragraph 1 and relevant figure
captions, as shown below:

“The initial flection amplitude of 12 mm was generated by using the elastic spring as a pre-load, which
corresponded to a pre-strain of 25 %.”

*COMMENT #3: There has been substantial work on lowering voltage in dielectric elastomers by increasing*
*dielectric constant, or reducing thickness. Early work on dielectrics was by Kofod I believe, and there has*
*been a lot since, while a number of groups have sought to make thin elastomer layers. This work has not been*
*properly summarized by the authors.*

**Response:** Following the reviewer’s suggestions, we have now summarized and discussed the earlier work on
increasing dielectric permittivity and reducing thickness of dielectric elastomers in the revised manuscript at
Page 3, Paragraph 2, as shown below:

“Numerous efforts have been devoted to increase the dielectric permittivity and mechanical flexibility to
generate a large actuation under relatively low driving voltages^{11-13,16-19}. For instance, the seminal work from
Kofod’s group enhanced the relative permittivity of the PDMS elastomer from 3.0 to 5.9 and decreased the
elastic modulus from 1900 to 550 kPa by grafting small molecules with high dipole moment to the elastomer
matrix, leading to significant improvement of their electromechanical performances¹⁹. In addition, the
reduction of the film thickness is an alternative method to improve the actuation performance²⁰⁻²³. For example,
Shea and his co-workers demonstrated that the actuation strain of 7.5 % could be generated with a 3 μm thick
film under a driving voltage of 245 V²⁰. By contrast, it required much higher driving voltage of 3.3 kV to
generate the same actuation strain with the 30 μm thick film. Despite these positive outcomes, thin film
actuators often require complicated fabrication processes and are associated with high prevalence of an
electromechanical instability²⁴.”

*COMMENT #4: The improved sensor response compared to PVC is well presented. It is unclear how this*
*performance compares to that of VHB and silicones.*

**Response:** To address the reviewer’s concern, we made PDMS and VHB 4910 based sensors and evaluated
their sensing performances, *i.e.* sensitivity and stability, under the exact same conditions as PVCg and
CEC/PVCg sensors. First, sensitivity tests showed that our CEC/PVCg sensors demonstrated the highest
sensitivity than other sensors, as shown in the following **Fig. R2**. Specifically, the sensitivity (*S*) of CEC/PVCg
sensors were 3.1-fold and 1.5-fold higher than PDMS and VHB 4910 sensors in the displacement ranges of 7-
14 mm, respectively. Second, the stability of sensing performance was evaluated over 1440 cycles of flection
(2.5 s per cycle and 60 min in total), as shown in **Fig. R3**. The profiles of relative capacitance over time (**Fig.**
**R3a-3d**) indicated the stable performance for CEC/PVCg and PDMS sensors while apparent shift over time
for PVCg and VHB 4910 sensors. The relative standard deviation (RSD) of relative capacitances over 1440
cycles was calculated to quantify the stability of their sensing performance (**Fig. R3e**). The results showed that
our CEC/PVCg and PDMS sensors displayed much lower RSD values (5.75 % and 3.67 %), *i.e.* higher stability,
than PVCg (9.84 %) and VHB 4910 (8.20 %) sensors. Altogether, our CEC/PVCg sensors demonstrated the
superior overall sensing performances regarding of high sensitivity and stability over currently existing PVCg,
PDMS, and VHB 4910 sensors. The new data has now been added as **Fig. 6c** and **6e** in the revised manuscript
and **Supplementary Note 5**. The relevant discussion has been added in the revised manuscript at Page 14,
Paragraphs 2 and 3, as shown below:

“To demonstrate the sensing application of the CEC/PVCg elastomers, the periodic strain driven by a linear
 reciprocating actuator was applied to the prepared DES devices, including CEC/PVCg, PVCg, PDMS, and
 VHB 4910-based sensors (Fig. 6a and Supplementary Information Fig. S17). The profiles and periods of
 the capacitance signals (output) that were generated from both PVCg and CEC/PVCg sensors were identical
 to the strain signals (input) (Fig. 6b and Supplementary Video S5), suggesting that the mechanical signal can
 be accurately converted into the electric signal by the prepared sensors (Supplementary Note 5). The
 CEC/PVCg sensors showed the fast response time, e.g. 1.0, 0.5, and 0.25 seconds under frequencies of 0.5-2.0
 403 Hz (Supplementary Information Fig. S19). Notably, the CEC/PVCg sensors generated a significantly higher
 signal/noise ratio, baseline capacitance and capacitance width (i.e. ΔC , difference between peak capacitance
 C and baseline capacitance C_0) than the PVCg sensors because of the higher permittivity of CEC/PVCg
 matrix³⁶. Moreover, the CEC/PVCg sensors showed the highest sensitivity (S) among four types of sensors
 that we studied here. For instance, the sensitivity of CEC/PVCg sensors was 3.1-fold, 1.5-fold, and 1.7-fold
 higher than PDMS, VHB 4910, and PVCg sensors, respectively in the displacement range of 7-14 mm (Fig.
 6c and Supplementary Information Fig. S18).”

“Notably, the capacitance generated by PVCg and VHB 4910 sensors showed an apparent drift over the
 recording time of 60 min (i.e. 1440 cycles) (Fig. 6d and Supplementary Information Fig. S20), which is in
 line with the previous report²⁸. By contrast, CEC/PVCg and PDMS sensors produced remarkably stable
 capacitance signals without visible drift over at least 60 min, which was resulted from the low viscoelasticity
 and the inhibition on the rearrangement of their polar functions by the multiple molecular interactions. The
 relative standard deviation (RSD) of capacitances over 1440 cycles was analyzed to quantify the stability of
 sensors (Fig. 6e). The results showed that our CEC/PVCg and PDMS sensors displayed much lower RSD
 values (5.75 % and 3.67 %) of relative capacitances, i.e. higher stability, than PVCg (9.84 %) and VHB 4910
 (8.2 %) sensors. Altogether, our CEC/PVCg sensors demonstrated the superior overall sensing performances
 regarding of high sensitivity and stability compared to existing PVCg, PDMS, and VHB 4910 sensors.”

**Fig. R2** Sensitivity tests. The relative capacitance ($C-C_0/C_0$) that was generated by PVCg, CEC/PVCg, PDMS,
 and VHB 4910 sensors as a function of the displacement. The tangential slope of the curve was defined as the
 sensitivity (S) of the sensors. The values of sensitivity in the displacement ranges of 7-14 mm were marked in
 the figure.

**Fig. R3** Sensing stability tests. Duration tests within 1440 flexion cycles ($T = 2.5$ s per cycle and 60 min in
 total) for the (a) PVCg, (b) CEC/PVCg, (c) VHB 4910, and (d) PDMS sensors. (e) The calculated relative
 standard deviation (RSD = standard deviation of relative capacitance / mean of relative capacitance) values of
 four types of sensors.

**COMMENT #5:** *What is the dielectric loss at typical operating frequencies/timescales - e.g. 1 Hz?*

**Response:** We thank the reviewer for bringing up this issue. In the original manuscript, we used an impedance
 analyzer (4294A, Agilent, USA), which is the most commonly used system in the literature [R1-R6], to
 measure the dielectric properties of our elastomers. The testing frequency range of this machine is $40 \sim 10^7$ Hz.
 Samples cannot be tested at 1 Hz with this machine. To answer the reviewer's question, we tested the dielectric
 properties of our elastomers using a new machine (Concept 80 system, Novocontrol, Germany) in the
 frequency range of $0.1 \sim 10^7$ Hz. The results showed that the dielectric losses of the CEC/PVCg elastomers with
 0, 1, 9, 17 wt% loading concentrations of CEC were 20.84, 22.98, 22.98, 39.67 @ 1 Hz, and 0.026, 0.031,
 0.031, 0.047 @ 1 kHz, respectively (**Fig. R4**). The dielectric loss is highly frequency dependent, *i.e.* decrease
 at higher frequency, which is in line with previous reports [R9-R12]. In addition, the values of dielectric losses
 at 1 kHz that were measured by traditional machine (4294A, Agilent) and the new machine (Concept 80 system)
 were different, *e.g.* 0.105 vs 0.031 at 1 kHz. Since most previous reports used the traditional machine (4294A,
 Agilent) to measure the dielectric properties at 1 kHz, we kept the original data that were acquired using
 traditional machine (4294A, Agilent) in the revised manuscript in order to have a better comparison to the
 previous literatures and avoid potential confusions.

**Fig. R4** Dielectric losses of the CEC/PVCg elastomers with 0, 1, 9, 17 wt% loading concentrations of CEC
 under frequencies of 0.1-10 MHz, that were measured using the new machine (Concept 80 system).

*COMMENT #6: It is claimed that the actuators and sensors show fast response. I don't see a frequency
 response or time response analysis, or comparison with other materials.*

**Our response:** Most of creep driven actuators, such as high viscoelastic PVCg, take 5-20 seconds to complete
 one actuation cycle [R13-R16]. By contrast, our CEC/PVCg actuators take only 0.2-1 second to finish one
 actuation cycle (as shown in the following **Fig. R5a**). The CEC/PVCg actuators presented a characteristic
 Maxwell field driven actuator, where the flexion amplitude decreases with the increase of frequency.

The capacitance sensors always showed a fast response from the previous literature. For instance,
 the capacitive strain sensor reported by Liu's group had a fast response time less than 140 ms [R17]. The
 flexible capacitive pressure sensor reported by Lee and Kim's group showed a response time of 0.578-1.04
 seconds [R18]. **Fig. R5b** recorded typical capacity/frequency curves for the CEC/PVCg elastomers under a
 constant flexion and different driving frequencies, including 0.5, 1.0, and 2.0 Hz. The typical response time
 of our CEC/PVCg sensors are 1.0, 0.5, and 0.25 seconds. In addition, the fast response of our CEC/PVCg
 sensors was also demonstrated by their faithful recording of the leg motion during fast running, where the
 response time is about 0.2 second, as shown in the **Fig. 6i in the manuscript** and **Supplementary Video S6**.

We have now added the relationship between the flexion displacement and the driven frequency in
 **Supplementary Note 4**, and relevant discussions in the revised manuscript at Page 11, Paragraph 3 and Page
 14, Paragraph 2 as shown below:

*“In addition, the amplitude of flexion displacement of our CEC/PVCg actuators was decreased with the
 increase of the driving frequency with fast response time (0.1-0.5 seconds) (**Supplementary Information Fig.
 S14**), which represented a characteristic electromechanical behavior of Maxwell field driven actuators. By
 contrast, it often took 5-20 seconds per cycle for creep-driven actuators, such as PVCg actuators²⁶.”*

*“The CEC/PVCg sensors showed the fast response time, e.g. 1.0, 0.5, and 0.25 seconds under frequencies of
 0.5-2.0 Hz (**Supplementary Information Fig. S19**).”*

**Fig. R5** Time response tests. (a) The flextion displacements that were generated by our CEC/PVCg (9 wt%
 CEC) actuators under different driving frequencies and 9.09 V/ μ m electrical field. (b) The output capacity of
 our CEC/PVCg (9 wt% CEC) sensors under different driving frequencies and 14.0 mm flextion displacement
 (input), which was generated from a commercial linear actuator.

**COMMENT #7:** *What are the potential drawbacks of the approach? Do we expect a greater temperature*
 *dependence? Is breakdown strength reduced? Can large actuation be achieved, by applying higher voltages*
 *(why did the authors stop at the 9 V/micron)?*

**Response:** We thank the reviewer for bringing up these important issues to discuss. A potential limitation of
 current device is the instinct low breakdown strength of the traditional PVCg, which was not improved by the
 addition of the CEC. The major reason is that the small plasticizer molecule *i.e.* DOP would percolate through
 the elastomer and generate the premature breakdown strength. The low breakdown strength limits the use of
 high driving voltage to achieve larger actuation. However, the introduction of CEC in the plasticized PVCg
 significantly increase their permittivity by 2.5 folds while maintaining the similar breakdown strength.
 Therefore, the CEC/PVCg can achieve large actuation under low driving electric field. Another limitation is
 the current film fabrication method, *i.e.* mold casting, which is very straightforward while producing a relative
 thick (\sim 470 μ m) film of the dielectric elastomer to ensure the uniformity of the film and large force output.
 The decrease of film thickness would be expected to lower the required driving electric field and increase the
 breakdown strength [R19,R20]. Thus, we are currently working on the fabrication of the thinner film, *e.g.* \sim 100
 μ m by using spray coating method, or down to \sim 50 μ m by using spinning coating method, in order to further
 improve the actuation performances of our devices under low driving electric field. We have now provided the
 discussion about the limitations of this study in the revised manuscript at Page17, Paragraph 3, as shown below:

*“One limitation of current devices is the intrinsic low breakdown strength, *i.e.* 21.79 V/ μ m, of the traditional*
 *plasticized PVCg, which has not been improved by the addition of CEC in this study. The low breakdown*
 *strength prevents the use of high driving electric field to achieve larger actuation. However, the introduction*
 *of CEC in the plasticized PVCg indeed significantly augmented their actuation by increasing their permittivity*
 *by 2.5 folds while maintaining the similar breakdown strength. Another limitation of this study is that the*
 *current film fabrication method, *i.e.* mold casting, produced a relative thick film (\sim 470 μ m) of DEs although*
 *it is very straightforward to use. It has been demonstrated that the decrease of film thickness would lower the*
 *required driving electric field and increase the breakdown strength^{51,52}. Therefore, we are currently working*

on the fabrication of the thinner film, e.g. $\sim 100 \mu\text{m}$ by using spray coating method, or down to $\sim 50 \mu\text{m}$ by
using spinning coating method, to further improve the actuation performances of DEAs under low driving
electric field.”

For the issue of temperature dependence of DEA/DES, the temperature is an important factor to
consider for its influences on the electromechanical performance, especially when the devices would be used
under environments with large temperature fluctuations. For example, Zhang’s study demonstrated that the
breakdown voltage of VHB 4910 decreased by increasing temperature [R21]. Michel *et al* and Vu-Conga *et al*
reported that the elastic moduli of DEs were significantly decreased at elevated temperature, facilitating the
generation of larger strains [R22,R23]. All the evaluations of this study were performed at room temperature.
The investigation of temperature dependence is needed when our DEA and DES are applied at different
environments in the future. To address the reviewer’s concern, we have added the discussions in the revised
manuscript at Page18, Paragraph 1, as shown below:

“In addition, the evaluation of all the actuation and sensing performances of our devices were evaluated under
room temperature. It has been reported that both breakdown strength and elastic modulus of DEs were sensitive
to variations of the temperature depending on materials⁵³. The temperature dependence needs to be evaluated
if our DEA and DES are used at different environments in the future.”

According to the following performance figures of merit, $S_z = \epsilon_r \epsilon_0 E^2 / Y$, a larger actuation strain would
be achieved by increasing the applied electric field. As demonstrated by our results (as shown in the following
**Fig. R6**), the PVCg-based actuators produced 1.7-fold higher area strain by increasing the driving voltage from
9.26 to 19.48 V/ μm . Existing DEAs often requires high driving electrical fields to achieve large strain, which,
however, could lead to the increased risks of polymer creep, current leakage, and electrical breakdown. In
addition, it also needs a bulky, high-voltage power supply system, hampering its wide-spread applications
[R24]. Therefore, a low driving voltage actuation is preferred for DEAs. In this study, the breakdown strength
of plasticized PVCg was about 20 V/ μm . The introduction of CEC and its loading concentration have limited
impacts on the breakdown strength while it dramatically increased permittivity. Thus, a relative low working
voltage of 9.09 V/ μm was selected in order to prolong the working life of the device, achieve a stable
performance while generating a desired large area strain [R25-R28]. Notably, such strain of 12.22 % was
achieved under a very small pre-strain of 25 %, which was negligible when compared to other pre-strains such
as 540 % and 400 % [R29, R30]. To address the reviewer’s comments, we have added the relevant discussions
in the revised manuscript at Page 8, Paragraph1, as shown below:

“Importantly, the breakdown strength was only slightly decreased by 10.4 % following the addition of CEC
and kept almost constant for CEC/PVCg elastomers with different loading concentrations of CEC (**Fig. 3c** and
**Supplementary Information Fig. S4**). The detected breakdown strength of CEC/PVCg elastomers with 9 wt%
CEC was 19.53 V/ μm . Thus, a much lower driving electric field of 9.09 V/ μm was used in all of experiments
in this study in order to prolong the working life of device and achieve a stable performance while generating
a desired large area strain.”

**Fig. R6** Actuation strains of the dopamine (PDA) coating ZnO particles hybrid PVCg (PDA@ZnO/PVCg)
 actuators under driving electrical fields of 9.26 and 19.48 V/μm.

*COMMENT #8: The sensor work is well presented. It is important to put this work put in context of other*
 *capacitive sensor work, and perhaps provide some detail in the supplementary. There are also commercial*
 *sensors (e.g. Stretchesense) to compare with.*

**Response:** We thank the reviewer for her/his great suggestion. We have now compared our CEC/PVCg sensors
 with three different types of currently existing sensors, as shown in the following **Table R1**. One of the major
 drawbacks of existing sensors is that they cannot achieve high sensitivity over a wide range of deformation.
 For instance, the commercial strain gauge sensor has a very high sensitivity, but it was limited to a narrow
 range of distance *e.g.* 0-0.12 mm [R31]. By contrast, Our CEC/PVCg sensors can achieve a high sensitivity of
 3.08 pF/mm over a much wider range of deformation, *i.e.* 0-14 mm. In addition, the relative high sensitivity
 of existing sensors has been heavily relying on the sophisticated structural design and micro/nano-
 manufacturing technologies [R32], such as micro-electromechanical systems (MEMS), resulting in the high
 complexity and high cost. By contrast, our CEC/PVCg sensor was fabricated by a simple mold-casting method
 and it costs ~ \$ 2.00 per sensor. To address the reviewer’s comment, we have added the following Table with
 new data as **Supplementary Table. S3** to compare our DEA against other commercially available sensors and
 relevant discussions in the revised manuscript at Page15, Paragraph 2, as shown below:

*“In addition, our CEC/PVCg sensors showed a high sensitivity in a wide range of deformation, which cannot*
 *be achieved with commercially available sensors (Supplementary Table S3). The high sensitivity of existing*
 *commercial sensors has been heavily relying on the sophisticated and complex design of their structures, which*
 *requires the costly and time-consuming micro-/nano-manufacturing techniques, such as micro-*
 *electromechanical systems (MEMS). By contrast, our CEC/PVCg sensors were fabricated by a very simple*
 *and highly accessible mold-casting method with an estimated cost of ~ \$ 2.00 per sensor.”*

Table R1 Comparison of magnetic, strain, and capacitive sensors

	Magnetic sensors ^[R31]	Strain gauge sensors ^[R33]	Capacitive sensor ^[R31]	CEC/PVCg Sensor
Stiffness	Rigid	Flexible	Flexible and stretchable	Flexible and stretchable
Range	100nm-70mm	0-0.12 mm	10nm-10 μ m	0-14 mm
Sensitivity	1.68 V/mm	Very high	0.038-5.3 pF/mm	3.08 pF/mm
Linearity (R^2)	0.9994	0.98-0.99	0.97-0.9975	0.988-0.992
Cost	Expensive	Expensive	Moderate	Cheap
Complexity	Complex	Complex	Complex	Simple

**References**

- [R1] You, I. et al. Artificial multimodal receptors based on ion relaxation dynamics. *Science* **370**, 961-965
(2020).
- [R2] Kim, P. et al. High energy density nanocomposites based on surface-modified BaTiO₃ and a ferroelectric
polymer. *ACS Nano* **3**, 2581-2592 (2009).
- [R3] Shao, J. et al. A novel high dielectric constant acrylic resin elastomer nanocomposite with pendant
oligoanilines. *Compos. B. Eng.* **176**, 107216 (2019).
- [R4] Fook, T. H. T., Jeon, J. Han. & Lee, P. S. Transparent flexible polymer actuator with enhanced Output
force enabled by conductive nanowires interlayer. *Adv. Mater. Technol.* **5**, 1900762 (2019).
- [R5] Xiong, Y. X. et al. A flexible, ultra-highly sensitive and stable capacitive pressure sensor with convex
microarrays for motion and health monitoring. *Nano Energy* **70**, 104436 (2020).
- [R6] Zhang, L., Song, F. L., Lin, X. & Wang, D. R. High-dielectric-permittivity silicone rubbers incorporated
with polydopamine-modified ceramics and their potential application as dielectric elastomer generator.
*Mater. Chem. Phys.* **241**, 122373 (2020).
- [R7] Shi, Y. et al. A processable, high-performance dielectric elastomer and multilayering process. *Science*
**377**, 228-232 (2022).
- [R8] Sheima, Y., Caspari, P. & Opris, D. M. Artificial muscles: dielectric elastomers responsive to low
voltages. *Macromol. Rapid Commun.* **40**, 1900205 (2019).
- [R9] Elkholly, M. M. & El-Deen, L. M. S. The dielectric properties of TeO₂-P₂O₅ glasses. *Mater. Chem. Phys.*
**65**, 192-196 (2000).
- [R10] Ning, N. Y. et al. Tailoring dielectric and actuated properties of elastomer composites by bioinspired
poly(dopamine) encapsulated graphene oxide. *ACS Appl. Mater. Interfaces* **7**, 10755-10762 (2015).
- [R11] Yang, D. et al. Enhancement of dielectric performance of polymer composites via constructing BaTiO₃-
poly(dopamine)-Ag nanoparticles through mussel-inspired surface functionalization. *ACS Omega* **3**,
14087-14096 (2018).
- [R12] Guo, Y. T. et al. Microstructure and dielectric properties of sub-micron hollow sphere
(Ba_{0.6}Sr_{0.4})TiO₃/PVDF composites. *IET Nanodielectrics* **2**, 135-141 (2019).
- [R13] Asaka, K. & Hashimoto, M. Effect of ionic liquids as additives for improving the performance of
plasticized PVC gel actuators. *Smart Mater. Struct.* **29**, 025003 (2020).
- [R14] Hwang, T., Frank, Z., Neubauer, J. & Kim, K. J. High-performance polyvinyl chloride gel artificial
muscle actuator with graphene oxide and plasticizer. *Sci. Rep.* **9**, 9658 (2019).
- [R15] Hirai, T., Zheng, J., Watanabe, M., Shirai, H. & Yamaguchi, M. Electroactive nonionic polymer gel
swift bending and crawling motion. *Mat. Res. Soc. Symp. Proc.* **600**, 267-272 (1999).
- [R16] Li, Y. & Hashimoto, M. PVC gel based artificial muscles: Characterizations and actuation modular
constructions. *Sens. Actuators A: Phys.* **233**, 246-258 (2015).
- [R17] Dong, T. Y., Gu, Y., Liu, T. & Pecht, M. Resistive and capacitive strain sensors based on customized
compliant electrode: Comparison and their wearable applications. *Sensor. Actuat. A Phys.* **326**, 112720
(2021).
- [R18] Beak, S. et al. Flexible piezocapacitive sensors based on wrinkled microstructures: toward low-cost
fabrication of pressure sensors over large areas. *RSC Adv.* **7**, 39420 (2017).
- [R19] Huang, J. S., Shian, S., Diebold, R. M., Suo, Z. G., & Clarke, D. R. The thickness and stretch dependence
of the electrical breakdown strength of an acrylic dielectric elastomer. *Appl. Phys. Lett.* **101**, 122905
(2012).

- [R20] Gatti, D. et al. The dielectric breakdown limit of silicone dielectric elastomer actuators. *Appl. Phys. Lett.*
**104**, 052905 (2014).
- [R21] Zhang, J. S. et al. Temperature effect on electromechanical properties of polyacrylic dielectric elastomer:
an experimental study. *Smart Mater. Struct.* **29**, 047002 (2020).
- [R22] Michel, S., Zhang, X. Q., Wissler, M., Löwe, C. & Kovacs, G. A comparison between silicone and
acrylic elastomers as dielectric materials in electroactive polymer actuators. *Polym. Int.* **59**, 391-399
(2010).
- [R23] Vu-Conga, T., Jean-Mistral, C. & Sylvester, A. New operating limits for applications with electroactive
elastomer: effect of the drift of the dielectric permittivity and the electrical breakdown *Proc. SPIE* **8687**,
86871S (2013).
- [R24] Chen, Z. Q. et al. Ultrasoft-yet-strong pentablock copolymer as dielectric elastomer highly responsive
to low voltages. *Chemical Engineering Journal. Chem. Eng. J.* **405**, 126634 (2021).
- [R25] Yin, L. J. et al. Soft, tough, and fast polyacrylate dielectric elastomer for non-magnetic motor. *Nat.*
*Commun.* **12**, 4517 (2021).
- [R26] Shankar, R., Ghosh, T. K. & Spontak, R. J. Electromechanical response of nanostructured polymer
systems with no mechanical pre-strain. *Macromol. Rapid Commun.* **28**, 1142-1147 (2007).
- [R27] Zhao, Y. et al. Remarkable electrically actuation performance in advanced acrylic-based dielectric
elastomers without pre-strain at very low driving electric field. *Polymer* **137**, 269-275 (2018).
- [R28] Yang, C. X., Gao, X. & Luo, Y. W. End-block-curing ABA triblock copolymer towards dielectric
elastomers with both high electro-mechanical performance and excellent mechanical properties. *Chem.*
*Eng. J.* **382**, 123037 (2020).
- [R29] Pelrine, R., Kornbluh, R., Pei, Q. B. & Joseph, J. High-speed electrically actuated elastomers with strain
greater than 100%. *Science* **287**, 836-845 (2000).
- [R30] Huang, J. S. et al. Giant, voltage-actuated deformation of a dielectric elastomer under dead load. *Appl.*
*Phys. Lett.* **100**, 041911 (2012).
- [R31] Stefanov, A. Z., Zivanov, L. D., Kistic, M. G. & Menicanin, A. B. Fully FFF-printed capacitive
displacement sensor based on graphene/PLA composite and thermoplastic elastomer filaments. *IEEE Sens.*
*J.* **22**, 10437-10445 (2022).
- [R32] Ruth, S. R. A. et al. Rational design of capacitive pressure sensors based on pyramidal microstructures
for specialized monitoring of biosignals. *Adv. Funct. Mater.* **30**, 1903100 (2020).
- [R33] Araromi, O. A. et al. Ultra-sensitive and resilient compliant strain gauges for soft machines. *Nature* **587**,
219 (2020).

**Response to Reviewer #3:**

We thank the reviewer for her/his constructive comment. We are delighted that the reviewer found “*the*
*physicochemical and electromechanical coupling property of the as-synthesized CEC/PVC were carefully*
*studied*” and “*This work improved the performance of traditional PVC.*” Here we address the comments and
technical questions raised by the reviewer with new experimental results and analyses.

**COMMENT #1:** *However, the demonstrated performance of actuation and sensing of the presented material*
*are not convincing to me. The paper cannot meet the standard of Nature Communications unless the authors*
*could clarify the advantage of the material compared to the existing material.*

**Response:** To address reviewer’s concern, we have prepared the actuators and sensors using the most
commonly used dielectric elastomers, *i.e.* PDMS and VHB 4910, and made a direct comparison of actuation
and sensing performances between theirs and our CEC/PVCg. For actuation performance, the generated area
strains of four types of actuators under the same pre-strain of 25 % and driving electric fields were first
measured. As shown in the following **Fig. R1a**, our CEC/PVCg actuators produced the largest area strains
among all actuators studied here. Our CEC/PVCg actuators showed 3.9-fold and 5-fold increase in area strain
as compared to PVCg and PDMS actuators, respectively. Specifically, the area strains were 1.46-12.22 % for
the CEC/PVCg actuators under the driving electrical fields of 5.45-9.09 V/ μ m, 0.81-3.15 % for the PVCg
actuators under 5.45-9.09 V/ μ m, 0.65-2.44 % for the PDMS actuators under 5.85-9.07 V/ μ m, while VHB 4910
actuators cannot be activated under the driving electrical field < 12.5 V/ μ m, *i.e.* strain of 0 %. The area strains
of VHB 4910 actuators were 0.72-3.45 % when the driving electrical field was further increased to 12.5-22.5
V/ μ m. Second, the actuation stability of actuators was evaluated by recording the displacements over 1000
cycles, *i.e.* 1000 seconds, and analyzing their relative displacement shifts (RDS) to quantify the stability. As
shown in the following **Fig. R1b**, the RDS values of VHB 4910, PDMS, PVCg, and CEC/PVCg actuators over
1000 cycles were 136.09 %, 5.70 %, 59.40 %, and 7.78 %, respectively. Our CEC/PVCg actuators showed a
very low shift of displacement, which was 94 % and 87 % reductions as compared to VHB 4910 and PVCg
actuators. Therefore, our CEC/PVCg actuators in this study produced the largest actuation strain with the
extremely low viscoelastic effects, demonstrating the significant improvements in actuation performances as
compared to the existing DEAs, such as PDMS and VHB 4190.

**Fig. R1** The actuation performances of CEC/PVCg actuators as compared to existing PVCg, PDMS, and VHB
4910 actuators as measured by area strain (mean values) and actuation stability through the quantification of
the relative displacement shift (RDS) over different number of cycles or seconds (1 second per cycle). The

RDSs were calculated by the following equation $RDS = \frac{|D - D_{creep}|}{D} \times 100\%$, where D is the amplitude of
 displacement and D_{creep} is the shift of the displacement.

For sensing performance (**Fig. R2**), the sensitivity of four types of sensors was first measured by
 calculating the slope of relative capacitance (output) as functions of displacement (input). As shown in **Fig.**
 **R2a**, our CEC/PVCg sensors demonstrated the highest sensitivity among all four types of sensors. For instance,
 the sensitivity (S) of CEC/PVCg sensors were 3.1-fold and 1.5-fold higher than PDMS and VHB 4910 sensors,
 respectively in the displacement ranges of 7-14 mm. Moreover, the stability of sensing performance was
 evaluated and quantified by analyzing the relative standard deviation (RSD) of relative capacitances over 1440
 cycles of flexion (2.5 s per cycle and 60 min in total), as shown in **Fig. R2b**. The results showed that our
 CEC/PVCg and PDMS sensors displayed much lower RSD values (5.75 % and 3.67 %), *i.e.* higher stability,
 than PVCg (9.84 %) and VHB 4910 (8.20 %) sensors. Altogether, our CEC/PVCg sensors demonstrated the
 superior overall sensing performances regarding of high sensitivity and stability over currently existing PVCg,
 PDMS, and VHB 4910 sensors.

**Fig. R2** (a) The relative capacitance $(C - C_0)/C_0$ that was generated by PVCg, CEC/PVCg, PDMS, and VHB
 4910 sensors as a function of the displacement. The tangential slope of the curve was defined as the sensitivity
 (S) of the sensors. The values of sensitivity in the displacement ranges of 7-14 mm were marked in the figure.
 (b) The relative standard deviation (RSD) of relative capacitances over 1440 cycles of flexion ($T = 2.5$ s per
 cycle and 60 min in total). $RSD = \text{standard deviation of relative capacitance change} / \text{mean of relative}$
 $\text{capacitance change}$.

 In summary, it is fair to make a conclusion that the CEC/PVCg dielectric elastomers developed in
 this study demonstrated the significant advantages over currently existing materials, *e.g.* PDMS and VHB 4910,
 for both actuation and sensing applications. We have now provided these new data and discussions in the
 revised manuscript at Page 11, Paragraph 3 as shown below:

“According to the strain model shown in **Supplementary Information Fig. S12**, the counted area strain
 generated by the CEC/PVCg actuators was 12.22 % (with 9 wt% CEC, under driving voltage of 9.09 V/ μm ,
 pre-strain of 25 %), which represents 3.9-fold increase as compared to the PVCg actuators (**Fig. 5b**). In addition,
 we prepared the commonly used PDMS and VHB-based actuators and measured their actuation strains under
 the same conditions as the comparison (**Fig. 5b**, **Supplementary Table S2**, and **Supplementary Information**
 **Fig. S15**). PDMS actuators produced strains of 0.65-2.44 % under electrical fields of 5.85-9.07 V/ μm . VHB
 4910 actuators could not be triggered, *i.e.* 0 % area strain, under electrical fields < 12.5 V/ μm and produced

only 0.72-3.45 % strains by further increasing electrical fields to 12.5-22.5 V/ μm . Therefore, our CEC/PVCg
actuators generated significantly larger actuation strains, *i.e.* > 5 times, than commonly used PDMS and VHB
4910 actuators, which was largely attributed to the augmentation of the electromechanical coupling sensitivity
k of CEC/PVCg (**Fig. 3e**). In addition, the amplitude of flexion displacement of our CEC/PVCg actuators was
decreased with the increase of the driving frequency with fast response time (0.1-0.5 seconds)
(**Supplementary Information Fig. S14**), which represented a characteristic electromechanical behavior of
Maxwell field driven actuators. By contrast, it often took 5-20 seconds per cycle for creep-driven actuators,
such as PVCg actuators²⁶.”

Page 12, Paragraph 2, as shown below:

“The displacements of the PVCg, CEC/PVCg, PDMS, and VHB 4910 actuators were measured and recorded
over actuation for 1000 cycles, *i.e.* 1000 seconds (**Fig. 5c** and **Supplementary Information Fig. S16**). The
CEC/PVCg and PDMS actuators produced remarkably stable displacement profiles. By contrast, apparent
displacement shifts were observed over time for PVCg and VHB 4910 actuators. Relative displacement shifts
(RDS) were calculated to quantify the viscoelastic effects (**Fig. 5d**). It was found that RDS values increased
with time, *i.e.* number of actuation cycles, for VHB 4910 and PVCg actuators, while remaining almost constant
for CEC/PVCg actuators. The relative shifts over 1000 cycles of CEC/PVCg actuators (7.78 % of RDS)
represented 87 % and 94 % reductions as compared to PVCg (59.40 % of RDS) and VHB 4910 actuators
(136.09 % of RDS). PDMS actuators (5.70 % of RDS) displayed similar viscoelastic drifts to CEC/PVCg
actuators.”

Page 14, Paragraph 2, as shown below:

“To demonstrate the sensing application of the CEC/PVCg elastomers, the periodic strain driven by a linear
reciprocating actuator was applied to the prepared DES devices, including CEC/PVCg, PVCg, PDMS, and
VHB 4910-based sensors (**Fig. 6a** and **Supplementary Information Fig. S17**). The profiles and periods of
the capacitance signals (output) that were generated from both PVCg and CEC/PVCg sensors were identical
to the strain signals (input) (**Fig. 6b** and **Supplementary Video S5**), suggesting that the mechanical signal can
be accurately converted into the electric signal by the prepared sensors (**Supplementary Note 5**). The
CEC/PVCg sensors showed the fast response time, *e.g.* 1.0, 0.5, and 0.25 seconds under frequencies of 0.5-2.0
780 Hz (**Supplementary Information Fig. S19**). Notably, the CEC/PVCg sensors generated a significantly higher
signal/noise ratio, baseline capacitance and capacitance width (*i.e.* ΔC , difference between peak capacitance
C and baseline capacitance C_0) than the PVCg sensors because of the higher permittivity of CEC/PVCg
matrix³⁶. Moreover, the CEC/PVCg sensors showed the highest sensitivity (S) among four types of sensors
that we studied here. For instance, the sensitivity of CEC/PVCg sensors was 3.1-fold, 1.5-fold, and 1.7-fold
higher than PDMS, VHB 4910, and PVCg sensors, respectively in the displacement range of 7-14 mm (**Fig.**
**6c** and **Supplementary Information Fig. S18**).”

Page 14, Paragraph 3, as shown below:

“Notably, the capacitance generated by PVCg and VHB 4910 sensors showed an apparent drift over the
recording time of 60 min (*i.e.* 1440 cycles) (**Fig. 6d** and **Supplementary Information Fig. S20**), which is in
line with the previous report²⁸. By contrast, CEC/PVCg and PDMS sensors produced remarkably stable
capacitance signals without visible drift over at least 60 min, which was resulted from the low viscoelasticity
and the inhibition on the rearrangement of their polar functions by the multiple molecular interactions. The

relative standard deviation (RSD) of capacitances over 1440 cycles was analyzed to quantify the stability of
sensors (**Fig. 6e**). The results showed that our CEC/PVCg and PDMS sensors displayed much lower RSD
values (5.75 % and 3.67 %) of relative capacitances, *i.e.* higher stability, than PVCg (9.84 %) and VHB 4910
(8.2 %) sensors. Altogether, our CEC/PVCg sensors demonstrated the superior overall sensing performances
regarding of high sensitivity and stability compared to existing PVCg, PDMS, and VHB 4910 sensors.”

**COMMENT #2:** *The strategy of increasing permittivity and reducing viscosity in the material synthesis part*
*sounds reasonable and interesting. However, if the material is aimed to serve for dielectric elastomer*
*applications, these two indexes are apparently not dominated material properties.*

**Response:** There are two major factors in materials properties of dielectric elastomers (DEs), *i.e.* permittivity
and mechanical properties, which are critical for their actuation and sensing applications according to the
following performance figures of merits:

$$\text{Actuation strain: } S_z = -\frac{\epsilon_0 \epsilon_r E^2}{Y} = -kE^2$$

$$\text{Sensing Capacitance : } C = \frac{\epsilon_0 \epsilon_r A}{d}$$

Where ϵ_0 and ϵ_r are the permittivity of free space and the relative permittivity of the elastomer
matrix, respectively, Y is the Young's modulus, E is the applied electrical field, d is the thickness of the matrix
film, k is electromechanical coupling sensitivity (ϵ/Y), and A is the area of electrodes.

Numerous studies have proposed and demonstrated that increasing dielectric permittivity and
mechanical flexibility of DE matrix are critical and very effective strategies to improve their actuation and
sensing performances [R1-R5]. For actuation applications, Pei *et al* reviewed materials innovations and
technological progress of DEAs, and concluded that “a high-performance DE should have sufficiently high
elastic strains, a large dielectric permittivity, high dielectric strength, and an actuation stability without
premature failure [R6,R7]”. For example, the acrylate copolymer containing 4 vol% Al nanoparticles has a
high dielectric permittivity of 8.4, which was increased by 78 % compared to the pure acrylate polymer, leading
to significant increase in their breakdown strength and actuation pressure [R8]. Recently, Opris and co-workers
successfully increased the dielectric permittivity of PDMS up to 18 by introducing the dipolar cyan group in
PDMS precursor, which leads to an actuation strain of 5.4 % at very low electric field of 3.2 V/ μm [R9]. For
sensing applications, the magnitude of output capacitance is proportional to the dielectric permittivity of the
elastomer [R5]. Increase of the DE permittivity would enlarge the magnitude of detected capacitance, resulting
in higher signal/noise ratio and sensitivity.

On the other hand, existing methods, such as the introduction of plasticizer into DEs, for lowering
the Young's moduli and increasing flexibility of DEs are often associated with the increase of viscoelastic
effects. High viscoelastic effects of DEs would result in evident mechanical loss, stress relaxation, and
viscoelastic hysteresis, leading to instability of output signals over time as well as delayed response [R10,R11].
For instance, the creep-driven PVCg actuators showed more frequent strain drifts and the delayed
electromechanical response as compared to actuators that are primarily driven by the Maxwell field [R12]. As
a sensor, the viscoelastic PVCg are often associated with a large signal shift of bulk permittivity and output
signals over time because of the random rearrangement of polar groups of PVC chain during stretching [R5].
However, such detrimental impacts of high viscoelasticity have been overlooked in the filed for long time.

In this study, it was found that the introduction of CEC into the commonly used PVCg elastomers
not only significantly increased the dielectric permittivity, leading to dramatic enhancement of actuation strain
and sensitivity, but also effectively mitigated their viscoelastic effects, resulting in highly stable actuation and

sensing performance over long time. Importantly, we have demonstrated the superior performance of our
CEC/PVCg-based DEA and DES in both actuation and sensing as compared to existing and commonly used
PDMS and VHB 4910-based actuators (**Fig. R1**) and sensors (**Fig. R2**). Altogether, we believe the concurrent
increase of dielectric permittivity and reduction of viscoelastic effects of DEs are critical and effective strategy
for the improvement of DE in actuation and sensing applications.

To address the reviewer's concern, we have now provided our new data about the direct comparison
of actuation and sensing performances between our CEC/PVCg and commonly used dielectric elastomers, *e.g.*
PDMS and VHB 4910, which directly demonstrated the efficiency of our strategy, in the revised manuscript
as **Fig. 5b**, **5d**, and **Fig. 6c**, **6e**. We have also stressed the importance of increasing dielectric permittivity and
reducing viscoelasticity in DEA and DES applications by adding more discussion in the revised manuscript,
at Page 3, Paragraph 2 and Page 4, Paragraph 2, as shown below:

“Numerous efforts have been devoted to increase the dielectric permittivity and mechanical flexibility to
generate a large actuation under relatively low driving voltages^{11-13,16-19}. For instance, the seminal work from
Kofod's group enhanced the relative permittivity of the PDMS elastomer from 3.0 to 5.9 and decreased the
elastic modulus from 1900 to 550 kPa by grafting small molecules with high dipole moment to the elastomer
matrix, leading to significant improvement of their electromechanical performances¹⁹. In addition, the
reduction of the film thickness is an alternative method to improve the actuation performance²⁰⁻²³. For example,
Shea and his co-workers demonstrated that the actuation strain of 7.5 % could be generated with a 3 μm thick
film under a driving voltage of 245 V²⁰. By contrast, it required much higher driving voltage of 3.3 kV to
generate the same actuation strain with the 30 μm thick film. Despite these positive outcomes, thin film
actuators often require complicated fabrication processes and are associated with high prevalence of an
electromechanical instability²⁴.”

“Notably, such viscoelastic effects are widely presented in other elastomers such as VHB^{12,32}, polyurethane
(PU)³³, and polyurethane acrylate (PUA)³⁴. Although the creep could be utilized to trigger different
mechanisms of deformation, such as bending, contracting, and crawling, it results in evident mechanical loss,
stress relaxation, and viscoelastic hysteresis, leading to instability of output signals over time as well as delayed
response³⁵. For instance, the creep-driven PVCg actuators²⁵ show more frequent jump of output signals and
the delayed electromechanical response as compared to actuators that are primarily driven by the Maxwell
force. The viscoelasticity of PVCg sensors often leads to a large drift of bulk permittivity and output signals
over time because of the random rearrangement of polar groups of PVC chain during stretching³⁶. However,
the viscoelastic effects of PVCg-based DEs have been largely over-looked. The mitigation of their viscoelastic
effects without compromising their electromechanical functions remains warranted.”

**COMMENT #3:** *As shown in Fig. 5, the actuation strain of the new material falls within the order of 10 %. A*
*lot of existing studies have shown that various of materials without careful optimization and complicated*
*synthesis can easily achieve this actuation level. Similar concerns apply for the sensing demo in this work.*

**Response:** For actuation applications, the generated actuation strains can have very large differences when
they are measured under different settings, *e.g.* different driving electric fields and pre-strains. The large area
strains often require high driving electric fields and large pre-strains. For instance, the commercial VHB
elastomer generated area strain of 215 % under very high electrical field of 239 V/μm and large pre-strain of
540 % [R13] while producing much smaller area strain of 34 % under electrical field of 70 V/μm and pre-
strain of 400 % [R14]. However, the high driving electric field (> 20 V/μm) could lead to the high risks of
current leakage [R15] and electrical breakdown [R16]. In addition, a high value of several kilovolts arises

safety issues and brings about the problem of using a bulky high-voltage power supply system [R17]. The
generation of large pre-strains can add extra complexity in the fabrication process of devices, reduce the
reproducibility, and increase risks of mechanical damage of the films. Therefore, the generation of large
actuation under low driving electric field and pre-strain is highly desirable for DEA application while it has
been a long-standing challenge. In this study, we evaluated the actuation performances of actuators under a
low driving electric field of $9.09 \text{ V}/\mu\text{m}$ and a small pre-strain of 25 %, which was negligible when compared
to other actuators [R13,R14]. To address the reviewer's concern, we have fabricated actuators using the most
commonly used dielectric elastomers, including PDMS and VHB 4910, and evaluated their actuation
performance under the same settings as our CEC/PVCg. As shown in **Fig. R1a**, our CEC/PVCg actuators
generated remarkably larger area strains (12.22 %) than PDMS (2.44 %) and VHB 4910 (no area strain, *i.e.*
0 %) actuators under low driving electric field of $9.09 \text{ V}/\mu\text{m}$ and small pre-strains of 25 %. Notably, VHB
4910 actuators generated only 3.45 % area strain by further increasing the driving electric field to $22.5 \text{ V}/\mu\text{m}$.
Moreover, our CEC/PVCg actuators exhibited a low mechanical loss and a high actuation stability due to the
significant mitigation of viscoelastic effects. As shown in the **Fig. R1b**, the VHB 4910 and pristine PVCg
actuators exhibited apparent shifts in displacement over 1000 actuation cycles, while the CEC/PVCg and
PDMS actuators did not show visible shift. The relative displacement shifts (RDS) were calculated over 1000
cycles to quantify their actuation stability. Our CEC/PVCg actuators showed the high actuation stability with
the low RDS value (7.78 % of RDS), which was 87 % and 94 % reductions as compared to PVCg (59.40 % of
RDS) and VHB 4910 actuators (136.09 % of RDS).

For sensing applications, we evaluated the sensitivity and stability of output signals of PDMS, VHB
4910, PVCg, and CEC/PVCg sensors under the same conditions to make a direct comparison. As shown in
**Fig. R2** and detailed description in our *responses to the reviewer's COMMENT #1*, our CEC/PVCg sensors
showed the highest sensitivity among all four types of sensors and very stable signal outputs over 1440 cycles
of flexion.

Altogether, it is fair to make a conclusion that the CEC/PVCg dielectric elastomers presented in this
study demonstrated the superior performances in both actuation and sensing over currently existing dielectric
elastomers, such as PDMS and VHB acrylic based materials. We have now provided the data of direct
comparisons in actuation and sensing performance among different materials in **Fig. 5** and **Fig. 6** in the revised
manuscript. We also provided the relevant discussions in the revised manuscript at Page 11, Paragraph 3, Page
12, Paragraph 2, Page 14, Paragraph 2, and Page 14, Paragraph 3, which has been shown in *our response to*
*the reviewer's COMMENT #1*.

**COMMENT #4:** *In the introduction, the authors pointed out that the VHB material, widely used in literature,*
*has a clear drawback in viscous property. However, the synthesized new material improves its viscous property*
*compared to VHB but significantly sacrifices its ability of large actuation, which, to the reviewer's opinion, is*
*not satisfactory for dielectric elastomer applications.*

**Response:** The large actuation strains of VHB materials as reported from previous literature are largely relying
on the use of very high driving electric fields and large pre-strains as applied in the evaluation. For instance,
the commercial VHB elastomer generated area strain of 215 % with very high electrical field of $239 \text{ V}/\mu\text{m}$ and
large pre-strain of 540 % [R13] while producing much smaller area strain of 34 % with electrical field of $70 \text{ V}/\mu\text{m}$
and pre-strain of 400 % [R14]. However, the generation of large actuation under low driving electric
fields and small pre-strains have been highly desirable for DEA applications [R18]. To make a fair comparison
between VHB 4910 and our CEC/PVCg, we have evaluated their actuation performances under the same
settings. As shown in **Fig. R1a**, our CEC/PVCg actuators showed significantly larger actuation strain of 12.22 %

than the VHB 4910 actuators with area strain of 0 % under a low driving electric field of 9.09 V/ μm and a
small pre-strain of 25 %. The VHB 4910 actuators produced an area strain of 3.45 % only by further increasing
the driving electric field up to 22.5 V/ μm . Moreover, our CEC/PVCg actuators exhibited a lower mechanical
loss and a higher actuation stability than VHB 4910 actuators due to the significant mitigation of their
viscoelastic effects. As shown in the **Fig. R1b**, the relative displacement shifts (RDS) were analyzed from the
recorded displacement profiles over 1000 cycles to quantify the actuation stability. VHB 4910 actuators
(136.09 % shifts) displayed 18 times higher relative shifts than our CEC/PVCg actuators (7.78 % shifts). In
addition, our CEC/PVCg sensors also showed higher sensitivity and produced much more stable sensing
signals than VHB 4910 sensors (**Fig. R2**). Therefore, we believe these new data fully support our conclusion
that the CEC/PVCg dielectric elastomer in this study exhibited superior performances in both actuation and
sensing over commercially available VHB acrylic elastomers.

We have now provided the data of direct comparisons in actuation and sensing performance among
different materials in **Fig. 5** and **Fig. 6** in the revised manuscript. We also provided the relevant discussions in
the revised manuscript at Page 11, Paragraph 3, Page 12, Paragraph 2, Page 14, Paragraph 2, and Page 14,
Paragraph 3, which has been shown in *our response to the reviewer's COMMENT #1*.

References

- [R1] Zhang Q. M. et al. An all-organic composite actuator material with a high dielectric constant. *Nature* **419**, 284-287 (2002).
- [R2] Carpi, F., Bauer, S. & De Rossi, D. Stretching dielectric elastomer performance. *Science* **330**, 1759-1761 (2010).
- [R3] Shi, Y. et al. A processable, high-performance dielectric elastomer and multilayering process. *Science* **377**, 228-232 (2022).
- [R4] Romasanta, L. J., Lopez-Manchado, M. A. & Verdejo, R. Increasing the performance of dielectric elastomer actuators: a review from the materials perspective. *Prog. Polym. Sci.* **51**, 188-211 (2015).
- [R5] Chhetry, A., Sharma, S., Yoon, H., Ko, S. & Park, J. Y. Enhanced sensitivity of capacitive pressure and strain sensor based on CaCu₃Ti₄O₁₂ wrapped hybrid sponge for wearable applications. *Adv. Funct. Mater.* **30**, 1910020 (2020).
- [R6] Qiu, Y., Zhang, E., Plamthottam, R. & Pei, Q. B. Dielectric elastomer artificial muscle: materials innovations and device explorations. *Acc. Chem. Res.* **52**, 2, 316-325 (2019).
- [R7] Chen, D. & Pei, Q. B. Electronic muscles and skins: a review of soft sensors and actuators. *Chem. Rev.* **117**, 11239-11268 (2017).
- [R8] Hu, W., Zhang, S. N., Niu, X. F., Liu, C. & Pei, Q. B. An aluminum nanoparticle-acrylate copolymer nanocomposite as a dielectric elastomer with a high dielectric constant. *J. Mater. Chem. C.* **2**, 1658 (2014).
- [R9] Szczepanski, J., Danner, P. M. & Opris D. M. Self-healable, self-repairable, and recyclable electrically responsive artificial muscles. *Adv. Sci.* **9**, 2202153 (2022).
- [R10] Liu, L. et al. Understanding reversible Maxwellian electroactuation in a 3M VHB dielectric elastomer with prestrain. *Polymer* **144**, 150-158 (2018).
- [R11] Tan, M. W. M., Thangavel, G. & Lee, P. S. Enhancing dynamic actuation performance of dielectric elastomer actuators by tuning viscoelastic effects with polar crosslinking. *NPG Asia Mater.* **11**, 62 (2019).
- [R12] Li, Y. & Hashimoto, M. PVC gel based artificial muscles: Characterizations and actuation modular constructions. *Sens. Actuators A: Phys.* **233**, 246-258 (2015).
- [R13] Pelrine, R., Kornbluh, R., Pei, Q. B. & Joseph, J. High-speed electrically actuated elastomers with strain greater than 100%. *Science* **287**, 836-845 (2000).
- [R14] Yin, L. J. et al. Soft, tough, and fast polyacrylate dielectric elastomer for non-magnetic motor. *Nat. Commun.* **12**, 4517 (2021).
- [R15] Topper, T., Osmani, B., Lorcher, S. & Muller, B. Leakage current, self-clearing and actuation efficiency of nanometer-thin, low-voltage dielectric elastomer transducers tailored by thermal evaporation. *Proc. SPIE* **10163**, 101631F (2017).
- [R16] Huang J. J. et al. Vinylsilane-rich silicone filled by polydimethylsiloxane encapsulated carbon black particles for dielectric elastomer actuator with enhanced out-of-plane actuations. *Chem. Eng. J.* **428**, 131354 (2022).
- [R17] Chen, Z. Q. et al. Ultrasoft-yet-strong pentablock copolymer as dielectric elastomer highly responsive to low voltages. *Chem. Eng. J.* **405**, 126634 (2021).
- [R18] Zhao, Y. et al. Remarkable electrically actuation performance in advanced acrylic-based dielectric elastomers without pre-strain at very low driving electric field. *Polymer* **137**, 269-275 (2018).

Reviewers' Comments:

Reviewer #1:

Remarks to the Author:

I have carefully read the response from the authors according to the suggestion of reviewer 2. all of 6 comments is explained or added some new contents to make a good understanding for potential readers. Though the dielectric material in this manuscript has also several disadvantages, the content in this revision is very rich and gives a new insight to learn the dielectric elastomer actuator. In this case, I suggest this revision with high quality can be considered to accept.

Reviewer #3:

Remarks to the Author:

The authors have conducted detailed experiments to compare the new PVC material with the existing materials. The results are encouraging. The paper can be recommended subject to the following minor revisions.

In the revised manuscript, the authors compared the actuation strain of PVC with VHB 4910 at the same electric field, which was called a "fair comparison", and stated the actuation performance is better, which I still do not fully agree. The reason is that as an dielectric actuation material, the energy conversion density is proportional to permittivity*electric field squared. The key material parameter is the electrical breakdown strength. As the authors cited, many literatures have made efforts in improving the permittivity and lowering modulus but not in breakdown strength. That's why VHB has such an outstanding high energy conversion density in actuation as well as in energy harvesting. A detailed review can be seen in Lu et al. Mechanics of dielectric elastomer structures A review, 2020. I would suggest that the authors frankly admit that the new material is better than VHB in low-voltage actuation and low hysteresis but not in a comprehensive manner. This advantage could be helpful in many applications that do not pursue large deformation or high energy density.

Responses to Reviewer #3:

COMMENT: A detailed review can be seen in Lu et al. Mechanics of dielectric elastomer structures A review, 2020. I would suggest that the authors frankly admit that the new material is better than VHB in low-voltage actuation and low hysteresis but not in a comprehensive manner. This advantage could be helpful in many applications that do not pursue large deformation or high energy density.

Response: We thank for and agreed with the reviewer's comments on the comprehensive comparison between our dielectric elastomer and commercial VHB4910. Following the reviewer's suggestion, we have revised our statements throughout the manuscript by stating that "Our CEC/PVCg actuators demonstrate superior actuation performances over the existing DE actuators *under low electrical fields*". We have added more discussion on the limitation of our material compared to existing VHB4910 and PDMS in the revised manuscript at Page 4, Paragraph 3, Page 11, Paragraph 3, Page 17, Paragraph 2, and Page 17, Paragraph 3 as shown below:

"In this study, we reported a valuable strategy to produce a PVCg-based dielectric elastomer with unprecedented properties, i.e. high permittivity, low viscoelasticity, and high flexibility, and further demonstrated its superior performances in both actuation and sensing applications, especially under low driving electrical fields."

"Therefore, our CEC/PVCg actuators generated significantly larger actuation strains, i.e. > 5 times, than commonly used PDMS and VHB 4910 actuators within < 22.5 V/ μ m electrical field, which was largely attributed to the augmentation of the electromechanical coupling sensitivity k of CEC/PVCg (Fig. 3e)."

"As a result, the CEC/PVCg actuators demonstrate superior actuation performance over the existing DE actuators, such as PDMS and VHB 4910, under low driving electrical fields."

"One limitation of current devices is the intrinsic low breakdown strength, i.e. 21.79 V/ μ m, of the traditional plasticized PVCg, as compared to the existing PDMS and VHB 4910. The use of higher driving electrical fields can offer high energy conversion density and energy harvesting with VHB 4910-based actuators⁵¹."